# Ode to an ODE

**Krzysztof Choromanski** [*]
Robotics at Google NY

**Jared Quincy Davis** [*]
DeepMind & Stanford University

**Valerii Likhosherstov** [*]
University of Cambridge

**Xingyou Song**
Google Brain

**Jean-Jacques Slotine**
Massachusetts Institute of Technology

**Jacob Varley**
Robotics at Google NY

**Honglak Lee**
Google Brain

**Adrian Weller**
University of Cambridge & The Alan Turing Institute

**Vikas Sindhwani**
Robotics at Google NY

## Abstract

We present a new paradigm for Neural ODE algorithms, called ODEtoODE, where time-dependent parameters of the main flow evolve according to a matrix flow on the orthogonal group $\mathcal{O}(d)$. This nested system of two flows, where the parameter-flow is constrained to lie on the compact manifold, provides stability and effectiveness of training and provably solves the gradient vanishing-explosion problem which is intrinsically related to training deep neural network architectures such as Neural ODEs. Consequently, it leads to better downstream models, as we show on the example of training reinforcement learning policies with evolution strategies, and in the supervised learning setting, by comparing with previous SOTA baselines. We provide strong convergence results for our proposed mechanism that are independent of the depth of the network, supporting our empirical studies. Our results show an intriguing connection between the theory of deep neural networks and the field of matrix flows on compact manifolds.

## 1 Introduction

Neural ODEs [13, 10, 27] are natural continuous extensions of deep neural network architectures, with the evolution of the intermediate activations governed by an ODE:

$$\frac{d\mathbf{x}_t}{dt} = f(\mathbf{x}_t, t, \theta), \tag{1}$$

parameterized by $\theta \in \mathbb{R}^n$ and where $f : \mathbb{R}^d \times \mathbb{R} \times \mathbb{R}^n \to \mathbb{R}^d$ is some nonlinear mapping defining dynamics. A solution to the above system with initial condition $\mathbf{x}_0$ is of the form:

$$\mathbf{x}_t = \mathbf{x}_0 + \int_{t_0}^{t} f(\mathbf{x}_s, s, \theta)ds,$$

and can be approximated with various numerical integration techniques such as Runge-Kutta or Euler methods [48]. The latter give rise to discretizations:

$$\mathbf{x}_{t+dt} = \mathbf{x}_t + f(\mathbf{x}_t, t, \theta)dt,$$

---

[*]equal contribution

that can be interpreted as discrete flows of ResNet-like computations [29] and establish a strong connection between the theory of deep neural networks and differential equations. This led to successful applications of Neural ODEs in machine learning. Those include in particular efficient time-continuous normalizing flows algorithms [11] avoiding the computation of the determinant of the Jacobian (a computational bottleneck for discrete variants), as well as modeling latent dynamics in time-series analysis, particularly useful for handling irregularly sampled data [44]. Parameterized neural ODEs can be efficiently trained via adjoint sensitivity method [13] and are characterized by constant memory cost, independent of the depth of the system, with different parameterizations encoding different weight sharing patterns across infinitesimally close layers.

Such Neural ODE constructions enable deeper models than would not otherwise be possible with a fixed computation budget; however, it has been noted that training instabilities and the problem of vanishing/exploding gradients can arise during the learning of very deep systems [43, 4, 23].

To resolve these challenges for discrete recurrent neural network architectures, several improvements relying on transition transformations encoded by orthogonal/Hermitian matrices were proposed [2, 33]. Orthogonal matrices, while coupled with certain classes of nonlinear mappings, provably preserve norms of the loss gradients during backpropagation through layers, yet they also incur potentially substantial Riemannian optimization costs [21, 14, 28, 1]. Fortunately, there exist several efficient parameterizations of the subgroups of the orthogonal group $\mathcal{O}(d)$ that, even though in principle reduce representational capacity, in practice produce high-quality models and bypass Riemannian optimization [36, 40, 34]. All these advances address discrete settings and thus it is natural to ask what can be done for continuous systems, which by definition are deep.

In this paper, we answer this question by presenting a new paradigm for Neural ODE algorithms, called ODEtoODE, where time-dependent parameters of the main flow evolve according to a matrix flow on the orthogonal group $\mathcal{O}(d)$. Such flows can be thought of as analogous to sequences of orthogonal matrices in discrete structured orthogonal models. By linking the theory of training Neural ODEs with the rich mathematical field of matrix flows on compact manifolds, we can reformulate the problem of finding efficient Neural ODE algorithms as a task of constructing expressive parameterized flows on the orthogonal group. We show in this paper on the example of orthogonal flows corresponding to the so-called *isospectral flows* [8, 9, 16], that such systems studied by mathematicians for centuries indeed help in training Neural ODE architectures (see: ISO-ODEtoODEs in Sec. 3.1). There is a voluminous mathematical literature on using isospectral flows as continuous systems that solve a variety of combinatorial problems including sorting, matching and more [8, 9], but to the best of our knowledge, we are the first who propose to learn them as stabilizers for training deep Neural ODEs.

Our proposed nested systems of flows, where the parameter-flow is constrained to lie on the compact manifold, provide stability and effectiveness of training, and provably solve the gradient vanishing/exploding problem for continuous systems. Consequently, they lead to better downstream models, as we show on a broad set of experiments (training reinforcement learning policies with evolution strategies and supervised learning). We support our empirical studies with strong convergence results, independent of the depth of the network. We are not the first to explore nested Neural ODE structure (see: [55]). Our novelty is in showing that such hierarchical architectures can be significantly improved by an entanglement with flows on compact manifolds.

To summarize, in this work we make the following contributions:

• We introduce new, explicit constructions of non-autonomous nested Neural ODEs (ODEtoODEs) where parameters are defined as rich functions of time evolving on compact matrix manifolds (Sec. 3). We present two main architectures: gated-ODEtoODEs and ISO-ODEtoODEs (Sec. 3.1).

• We establish convergence results for ODEtoODEs on a broad class of Lipschitz-continuous objective functions, in particular in the challenging reinforcement learning setting (Sec. 4.2).

• We then use the above framework to outperform previous Neural ODE variants and baseline architectures on RL tasks from $\mathrm{OpenAI\ Gym}$ and the $\mathrm{DeepMind\ Control\ Suite}$, and simultaneously to yield strong results on image classification tasks. To the best of our knowledge, we are the first to show that well-designed Neural ODE models with a compact number of parameters make them good candidates for training reinforcement learning policies via evolutionary strategies (ES) [15].

All proofs are given in the Appendix. We conclude in Sec. 6 and discuss broader impact in Sec. 7.

## 2 Related work

Our work lies in the intersection of several fields: the theory of Lie groups, Riemannian geometry and deep neural systems. We provide an overview of the related literature below.

**Matrix flows.** Differential equations (DEs) on manifolds lie at the heart of modern differential geometry [17] which is a key ingredient to understanding structured optimization for stabilizing neural network training [14]. In our work, we consider in particular matrix gradient flows on $\mathcal{O}(d)$ that are solutions to trainable matrix DEs. In order to efficiently compute these flows, we leverage the theory of compact Lie groups that are compact smooth manifolds equipped with rich algebraic structure [35, 7, 42]. For efficient inference, we apply local linearizations of $\mathcal{O}(d)$ via its Lie algebra (skew-symmetric matrices vector spaces) and exponential mappings [50] (see: Sec. 3).

**Hypernetworks.** An idea to use one neural network (hypernetwork) to provide weights for another network [26] can be elegantly adopted to the nested Neural ODE setting, where [55] proposed to construct time-dependent parameters of one Neural ODE as a result of concurrently evolving another Neural ODE. While helpful in practice, this is insufficient to fully resolve the challenge of vanishing and exploding gradients. We expand upon this idea in our gated-ODEtoODE network, by constraining the hypernetwork to produce skew-symmetric matrices that are then translated to those from the orthogonal group $\mathcal{O}(d)$ via the aforementioned exponential mapping.

**Stable Neural ODEs.** On the line of work for stabilizing training of Neural ODEs, [22] proposes regularization based on optimal transport, while [37] adds Gaussian noise into the ODE equation, turning it into a stochastic dynamical system. [20] lifts the ODE to a higher dimensional space to prevent trajectory intersections, while [25, 39] add additional terms into the ODE, inspired by well known physical dynamical systems such as Hamiltonian dynamics.

**Orthogonal systems.** For classical non-linear networks, a broad class of methods which improve stability and generalization involve orthogonality constraints on the layer weights. Such methods range from merely orthogonal initialization [46, 32] and orthogonal regularization [3, 53] to methods which completely constrain weight matrices to the orthogonal manifold throughout training using Riemannian optimization [35], by projecting the unconstrained gradient to the orthogonal group $\mathcal{O}(d)$ [47]. Such methods have been highly useful in preventing exploding/vanishing gradients caused by long term dependencies in recurrent neural networks (RNNs) [30, 31]. [52, 12] note that *dynamical isometry* can be preserved by orthogonality, which allows training networks of very large depth.

## 3 Improving Neural ODEs via flows on compact matrix manifolds

Our core ODEtoODE architecture is the following nested Neural ODE system:

$$\begin{cases} \dot{\mathbf{x}}_t = f(\mathbf{W}_t \mathbf{x}_t) \\ \dot{\mathbf{W}}_t = \mathbf{W}_t b_\psi(t, \mathbf{W}_t) \end{cases} \tag{2}$$

for some function $f : \mathbb{R} \to \mathbb{R}$ (applied elementwise), and a parameterized function: $b_\psi : \mathbb{R} \times \mathbb{R}^{d \times d} \to \mathrm{Skew}(d)$, where $\mathrm{Skew}(d)$ stands for the vector space of skew-symmetric (antisymmetric) matrices in $\mathbb{R}^{d \times d}$. We take $\mathbf{W}_0 \in \mathcal{O}(d)$, where the *orthogonal group* $\mathcal{O}(d)$ is defined as: $\mathcal{O}(d) = \{\mathbf{M} \in \mathbb{R}^{d \times d} : \mathbf{M}^\top \mathbf{M} = \mathbf{I}_d\}$. It can be proven [35] that under these conditions $\mathbf{W}_t \in \mathcal{O}(d)$ for every $t \geq 0$. In principle ODEtoODE can exploit arbitrary compact matrix manifold $\Sigma$ (with $b_\psi$ modified respectively so that second equation in Formula 2 represents on-manifold flow), yet in practice we find that taking $\Sigma = \mathcal{O}(d)$ is the most effective. Furthermore, for $\Sigma = \mathcal{O}(d)$, the structure of $b_\psi$ is particularly simple, it is only required to output skew-symmetric matrices. Schematic representation of the ODEtoODE architecture is given in Fig. 1. We take as $f$ a function that is non-differentiable in at most finite number of its inputs and such that $|f'(x)| = 1$ on all others (e.g. $f(x) = |x|$).

### 3.1 Shaping ODEtoODEs

An ODEtoODE is defined by: $\mathbf{x}_0$, $\mathbf{W}_0$ (initial conditions) and $b_\psi$. Vector $\mathbf{x}_0$ encodes input data e.g. state of an RL agent (Sec. 5.1) or an image (Sec. 5.2). Initial point $\mathbf{W}_0$ of the matrix flow can be either learned (see: Sec. 3.2) or sampled upfront uniformly at random from Haar measure on $\mathcal{O}(d)$. We present two different parameterizations of $b_\psi$, leading to two different classes of ODEtoODEs:

**ISO-ODEtoODEs:** Those leverage a popular family of the *isospectral flows* (i.e. flows preserving matrix spectrum), called *double-bracket flows*, and given as: $\dot{\mathbf{H}}_t = [\mathbf{H}_t, [\mathbf{H}_t, \mathbf{N}]]$, where:

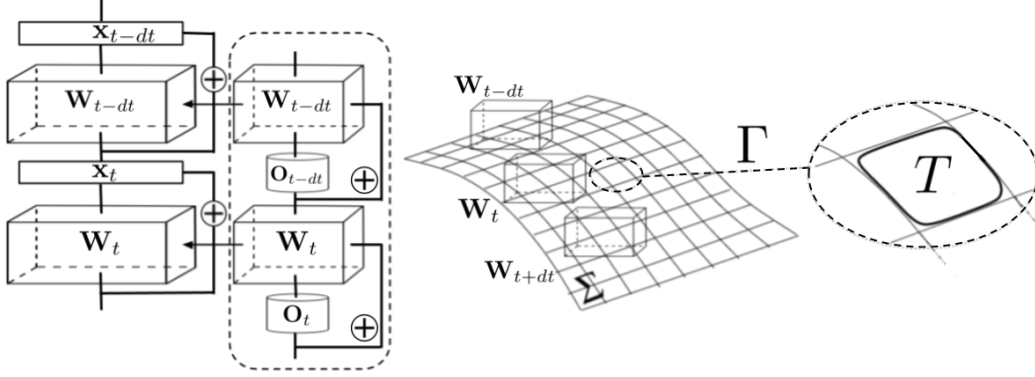

Figure 1: Schematic representation of the discretized ODEtoODE architecture. On the left: nested ODE structure with parameters of the main flow being fed with the output of the matrix flow (inside dashed contour). On the right: the matrix flow evolves on the compact manifold $\Sigma$, locally linearized by vector space $T$ and with mapping $\Gamma$ encoding this linearization ($\mathbf{O}_t = b_\psi(t, \mathbf{W}_t)$).

$\mathbf{Q} \stackrel{\text{def}}{=} \mathbf{H}_0, \mathbf{N} \in \text{Sym}(d) \subseteq \mathbb{R}^{d \times d}$, $\text{Sym}(d)$ stands for the class of symmetric matrices in $\mathbb{R}^{d \times d}$ and $[]$ denotes the *Lie bracket*: $[\mathbf{A}, \mathbf{B}] \stackrel{\text{def}}{=} \mathbf{AB} - \mathbf{BA}$. Double-bracket flows with customized parameter-matrices $\mathbf{Q}, \mathbf{N}$ can be used to solve various combinatorial problems such as sorting, matching, etc. [9]. It can be shown that $\mathbf{H}_t$ is similar to $\mathbf{H}_0$ for every $t \geq 0$, thus we can write: $\mathbf{H}_t = \mathbf{W}_t \mathbf{H}_0 \mathbf{W}_t^{-1}$, where $\mathbf{W}_t \in \mathcal{O}(d)$. The corresponding flow on $\mathcal{O}(d)$ has the form: $\dot{\mathbf{W}}_t = \mathbf{W}_t[(\mathbf{W}_t)^\top \mathbf{Q} \mathbf{W}_t, \mathbf{N}]$ and we take: $b_\psi^{\text{iso}}(t, \mathbf{W}_t) \stackrel{\text{def}}{=} [(\mathbf{W}_t)^\top \mathbf{Q} \mathbf{W}_t, \mathbf{N}]$, where $\psi = (\mathbf{Q}, \mathbf{N})$ are learnable symmetric matrices. It is easy to see that $b_\psi$ outputs skew-symmetric matrices.

**Gated-ODEtoODEs:** In this setting, inspired by [10, 55], we simply take $b_\psi^{\text{gated}} = \sum_{i=1}^{d} a_i B_\psi^i$, where $d$ stands for the number of gates, $a = (a_1, ..., a_d)$ are learnable coefficients and $B_\psi^i \stackrel{\text{def}}{=} f_i^{\psi^i} - (f_i^{\psi^i})^\top$. Here $\{f_i^{\psi^i}\}_{i=1,...,d}$ are outputs of neural networks parameterized by $\psi^i$s and producing unstructured matrices and $\psi = \text{concat}(a, \psi^1, ..., \psi^d)$. As above, $b_\psi^{\text{gated}}$ outputs matrices in $\text{Skew}(d)$.

### 3.2 Learning and executing ODEtoODEs

Note that most of the parameters of ODEtoODE from Formula 2 are unstructured and can be trained via standard methods. To train an initial orthogonal matrix $\mathbf{W}_0$, we apply tools from the theory of Lie groups and Lie algebras [35]. First, we notice that $\mathcal{O}(d)$ can be locally linearized via skew-symmetric matrices $\text{Skew}(d)$ (i.e. a tangent vector space $T_\mathbf{W}$ to $\mathcal{O}(d)$ in $\mathbf{W} \in \mathcal{O}(d)$ is of the form: $T_\mathbf{W} = \mathbf{W} \cdot \text{Sk}(d)$). We then compute Riemannian gradients which are projections of unstructured ones onto tangent spaces. For the inference on $\mathcal{O}(d)$, we apply exponential mapping $\Gamma(\mathbf{W}) \stackrel{\text{def}}{=} \exp(\eta b_\psi(t, \mathbf{W}))$, where $\eta$ is the step size, leading to the discretization of the form: $\mathbf{W}_{t+dt} = \mathbf{W}_t \Gamma(\mathbf{W}_t)$ (see: Fig. 1).

## 4 Theoretical results

### 4.1 ODEtoODEs solve gradient vanishing/explosion problem

First we show that ODEtoODEs do not suffer from the gradient vanishing/explosion problem, i.e. the norms of loss gradients with respect to intermediate activations $\mathbf{x}_t$ do not grow/vanish exponentially while backpropagating through time (through *infinitesimally close* layers).

**Lemma 4.1** (ODEtoODES for gradient stabilization). *Consider a Neural ODE on time interval $[0, T]$ and given by Formula 2. Let $\mathcal{L} = \mathcal{L}(\mathbf{x}_T)$ be a differentiable loss function. The following holds for any $t \in [0, 1]$, where $e = 2.71828...$ is Euler constant:*

$$\frac{1}{e} \|\frac{\partial \mathcal{L}}{\partial \mathbf{x}_T}\|_2 \leq \|\frac{\partial \mathcal{L}}{\partial \mathbf{x}_t}\|_2 \leq e \|\frac{\partial \mathcal{L}}{\partial \mathbf{x}_T}\|_2. \tag{3}$$

## 4.2 ODEtoODEs for training reinforcement learning policies with ES

Here we show strong convergence guarantees for ODEtoODE. For the clarity of the exposition, we consider ISO-ODEtoODE, even though similar results can be obtained for gated-ODEtoODE. We focus on training RL policies with ES [45], which is mathematically more challenging to analyze than supervised setting, since it requires applying ODEtoODE several times throughout the rollout. Analogous results can be obtained for the conceptually simpler supervised setting.

Let $\text{env} : \mathbb{R}^d \times \mathbb{R}^m \to \mathbb{R}^d \times \mathbb{R}$ be an "environment" function such that it gets current state $\mathbf{s}_k \in \mathbb{R}^d$ and action encoded as a vector $\mathbf{a}_k \in \mathbb{R}^m$ as its input and outputs the next state $\mathbf{s}_{k+1}$ and the next score value $l_{k+1} \in \mathbb{R}$: $(\mathbf{s}_{k+1}, l_{k+1}) = \text{env}(\mathbf{s}_k, \mathbf{a}_k)$. We treat $\text{env}$ as a "black-box" function, which means that we evaluate its value for any given input, but we don't have access to $\text{env}$'s gradients. An overall score $L$ is a sum of all per-step scores $l_k$ for the whole rollout from state $\mathbf{s}_0$ to state $\mathbf{s}_K$:

$$L(\mathbf{s}_0, \mathbf{a}_0, \dots, \mathbf{a}_{K-1}) = \sum_{k=1}^{K} l_k, \quad \forall k \in \{1, \dots, K\} : (\mathbf{s}_k, l_k) = \text{env}(\mathbf{s}_{k-1}, \mathbf{a}_{k-1}). \quad (4)$$

We assume that $\mathbf{s}_0$ is deterministic and known. The goal is to train a policy function $g_\theta : \mathbb{R}^d \to \mathbb{R}^m$ which, for the current state vector $\mathbf{s} \in \mathbb{R}^d$, returns an action vector $\mathbf{a} = g_\theta(\mathbf{s})$ where $\theta$ are trained parameters. By *Stiefel manifold* $\mathcal{ST}(d_1, d_2)$ we denote a nonsquare extension of orthogonal matrices: if $d_1 \geq d_2$ then $\mathcal{ST}(d_1, d_2) = \{\mathbf{\Omega} \in \mathbb{R}^{d_1 \times d_2} \,|\, \mathbf{\Omega}^\top \mathbf{\Omega} = I\}$, otherwise $\mathcal{ST}(d_1, d_2) = \{\mathbf{\Omega} \in \mathbb{R}^{d_1 \times d_2} \,|\, \mathbf{\Omega}\mathbf{\Omega}^\top = I\}$. We define $g_\theta$ as an ODEtoODE with Euler discretization given below:

$$g_\theta(\mathbf{s}) = \mathbf{\Omega}_2 \mathbf{x}_N, \quad \mathbf{x}_0 = \mathbf{\Omega}_1 \mathbf{s}, \quad \forall i \in \{1, \dots, N\} : \mathbf{x}_i = \mathbf{x}_{i-1} + \frac{1}{N} f(\mathbf{W}_i \mathbf{x}_{i-1} + \mathbf{b}) \quad (5)$$

$$\mathbf{W}_i = \mathbf{G}_{i-1} \exp\left( \frac{1}{N} (\mathbf{W}_{i-1}^\top \mathbf{Q} \mathbf{W}_{i-1} \mathbf{N} - \mathbf{N} \mathbf{W}_{i-1}^\top \mathbf{Q} \mathbf{W}_{i-1}) \right) \in \mathcal{O}(d), \quad (6)$$

$$\theta = \{\mathbf{\Omega}_1 \in \mathcal{ST}(n, d), \mathbf{\Omega}_2 \in \mathcal{ST}(m, n), \mathbf{b} \in \mathbb{R}^n, \mathbf{N} \in \mathbb{S}(n), \mathbf{Q} \in \mathbb{S}(n), \mathbf{W}_0 \in \mathcal{O}(n)\} \in \mathbb{D}$$

where by $\mathbb{D}$ we denote $\theta$'s domain: $\mathbb{D} = \mathcal{ST}(n, d) \times \mathcal{ST}(m, n) \times \mathbb{R}^n \times \mathbb{S}(n) \times \mathbb{S}(n) \times \mathcal{O}(n)$. Define a final objective $F : \mathbb{D} \to \mathbb{R}$ to maximize as

$$F(\theta) = L(\mathbf{s}_0, \mathbf{a}_0, \dots, \mathbf{a}_{K-1}), \quad \forall k \in \{1, \dots, K\} : \mathbf{a}_{k-1} = g_\theta(\mathbf{s}_{k-1}). \quad (7)$$

For convenience, instead of $(\mathbf{s}_{out}, l) = \text{env}(\mathbf{s}_{in}, \mathbf{a})$ we will write $\mathbf{s}_{out} = \text{env}^{(1)}(\mathbf{s}_{in}, \mathbf{a})$ and $l = \text{env}^{(2)}(\mathbf{s}_{in}, \mathbf{a})$. In our subsequent analysis we will use a not quite limiting Lipschitz-continuity assumption on $\text{env}$ which intuitively means that a small perturbation of $\text{env}$'s input leads to a small perturbation of its output. In addition to that, we assume that per-step loss $l_k = \text{env}^{(2)}(\mathbf{s}_{k-1}, \mathbf{a}_{k-1})$ is uniformly bounded.

**Assumption 4.2.** *There exist* $M, L_1, L_2 > 0$ *such that for any* $\mathbf{s}', \mathbf{s}'' \in \mathbb{R}^d, \mathbf{a}', \mathbf{a}'' \in \mathbb{R}^m$ *it holds:*

$$|\text{env}^{(2)}(\mathbf{s}', \mathbf{a}')| \leq M, \quad \|\text{env}^{(1)}(\mathbf{s}', \mathbf{a}') - \text{env}^{(1)}(\mathbf{s}'', \mathbf{a}'')\|_2 \leq L_1 \delta,$$

$$|\text{env}^{(2)}(\mathbf{s}', \mathbf{a}') - \text{env}^{(2)}(\mathbf{s}'', \mathbf{a}'')| \leq L_2 \delta, \quad \delta = \|\mathbf{s}' - \mathbf{s}''\|_2 + \|\mathbf{a}' - \mathbf{a}''\|_2.$$

**Assumption 4.3.** $f(\cdot)$ *is Lipschitz-continuous with a Lipschitz constant 1:* $\forall x', x'' \in \mathbb{R} : |f(x') - f(x'')| \leq |x' - x''|$. *In addition to that,* $f(0) = 0$.

Instead of optimizing $F(\theta)$ directly, we opt for optimization of a Gaussian-smoothed proxi $F_\sigma(\theta)$:

$$F_\sigma(\theta) = \mathbb{E}_{\epsilon \sim \mathcal{N}(0, I)} F(\theta + \sigma \epsilon)$$

where $\sigma > 0$. Gradient of $F_\sigma(\theta)$ has a form:

$$\nabla F_\sigma(\theta) = \frac{1}{\sigma} \mathbb{E}_{\epsilon \sim \mathcal{N}(0, I)} F(\theta + \sigma \epsilon) \epsilon,$$

which suggests an unbiased stochastic approximation such that it only requires to evaluate $F(\theta)$, i.e. no access to $F$'s gradient is needed, where $v$ is a chosen natural number:

$$\widetilde{\nabla} F_\sigma(\theta) = \frac{1}{\sigma v} \sum_{w=1}^{v} F(\theta + \sigma \epsilon_w) \epsilon_w, \quad \epsilon_1, \dots, \epsilon_v \sim \text{i.i.d. } \mathcal{N}(0, I),$$

By defining a corresponding metric, $\mathbb{D}$ becomes a product of Riemannian manifolds and a Riemannian manifold itself. By using a standard Euclidean metric, we consider a Riemannian gradient of $F_\sigma(\theta)$:

$$\nabla_\mathcal{R} F_\sigma(\theta) = \{\nabla_{\mathbf{\Omega}_1}\mathbf{\Omega}_1^\top - \mathbf{\Omega}_1\nabla_{\mathbf{\Omega}_1}^\top, \nabla_{\mathbf{\Omega}_2}\mathbf{\Omega}_2^\top - \mathbf{\Omega}_2\nabla_{\mathbf{\Omega}_2}^\top, \nabla_\mathbf{b}, \frac{1}{2}(\nabla_\mathbf{N} + \nabla_\mathbf{N}^\top), \frac{1}{2}(\nabla_\mathbf{Q} + \nabla_\mathbf{Q}^\top), \quad (8)$$

$$\nabla_{\mathbf{W}_0}\mathbf{W}_0^\top - \mathbf{W}_0\nabla_{\mathbf{W}_0}^\top\}, \text{ where } \{\nabla_{\mathbf{\Omega}_1}, \nabla_{\mathbf{\Omega}_2}, \nabla_\mathbf{b}, \nabla_\mathbf{N}, \nabla_\mathbf{Q}, \nabla_{\mathbf{W}_0}\} = \nabla F_\sigma(\theta). \quad (9)$$

We use Stochastic Riemannian Gradient Descent [6] to maximize $F(\theta)$. Stochastic Riemannian gradient estimate $\widetilde{\nabla}_\mathcal{R} F_\sigma(\theta)$ is obtained by substituting $\nabla \to \widetilde{\nabla}$ in (8-9). The following Theorem proves that SRGD is converging to a stationary point of the maximization objective $F(\theta)$ with rate $O(\tau^{-0.5+\epsilon})$ for any $\epsilon > 0$. Moreover, the constant hidden in $O(\tau^{-0.5+\epsilon})$ rate estimate doesn't depend on the length $N$ of the rollout.

**Theorem 1.** *Consider a sequence* $\{\theta^{(\tau)} = \{\{\mathbf{\Omega}_1^{(\tau)}, \mathbf{\Omega}_2^{(\tau)}, \mathbf{b}^{(\tau)}, \mathbf{N}^{(\tau)}, \mathbf{Q}^{(\tau)}, \mathbf{W}_0^{(\tau)}\} \in \mathbb{D}\}_{\tau=0}^\infty$ *where* $\theta^{(0)}$ *is deterministic and fixed and for each* $\tau > 0$:

$$\mathbf{\Omega}_1^{(\tau)} = \exp(\alpha_\tau \widetilde{\nabla}_{\mathcal{R},\mathbf{\Omega}_1}^{(\tau)})\mathbf{\Omega}_1^{(\tau-1)}, \quad \mathbf{\Omega}_2^{(\tau)} = \exp(\alpha_\tau \widetilde{\nabla}_{\mathcal{R},\mathbf{\Omega}_2}^{(\tau)})\mathbf{\Omega}_2^{(\tau-1)}, \quad \mathbf{b}^{(\tau)} = \mathbf{b}^{(\tau-1)} + \alpha_\tau \widetilde{\nabla}_{\mathcal{R},\mathbf{b}}^{(\tau)},$$

$$\mathbf{N}^{(\tau)} = \mathbf{N}^{(\tau-1)} + \alpha_\tau \widetilde{\nabla}_{\mathcal{R},\mathbf{N}}^{(\tau)}, \quad \mathbf{Q}^{(\tau)} = \mathbf{Q}^{(\tau-1)} + \alpha_\tau \widetilde{\nabla}_{\mathcal{R},\mathbf{Q}}^{(\tau)}, \quad \mathbf{W}_0^{(\tau)} = \exp(\alpha_\tau \widetilde{\nabla}_{\mathcal{R},\mathbf{W}_0}^{(\tau)})\mathbf{W}_0^{(\tau-1)},$$

$$\{\widetilde{\nabla}_{\mathcal{R},\mathbf{\Omega}_1}^{(\tau)}, \widetilde{\nabla}_{\mathcal{R},\mathbf{\Omega}_2}^{(\tau)}, \widetilde{\nabla}_{\mathcal{R},\mathbf{b}}^{(\tau)}, \widetilde{\nabla}_{\mathcal{R},\mathbf{N}}^{(\tau)}, \widetilde{\nabla}_{\mathcal{R},\mathbf{Q}}^{(\tau)}, \widetilde{\nabla}_{\mathcal{R},\mathbf{W}_0}^{(\tau)}\} = \widetilde{\nabla}_\mathcal{R} F_\sigma(\theta^{(\tau)}), \quad \alpha_\tau = \tau^{-0.5}.$$

*Then* $\min_{0\leq\tau'<\tau} \mathbb{E}[\|\nabla_\mathcal{R} F_\sigma(\theta^{(\tau')})\|_2^2 | \mathcal{F}_{\tau,D,D_b}] \leq \mathcal{E} \cdot \tau^{-0.5+\epsilon}$ *for any* $\epsilon > 0$ *where* $\mathcal{E}$ *doesn't depend on* $N$ *and* $D, D_b > 0$ *are constants and* $\mathcal{F}_{\tau,D,D_b}$ *denotes a condition that for all* $\tau' \leq \tau$ *it holds* $\|\mathbf{N}^{(\tau')}\|_2, \|\mathbf{Q}^{(\tau')}\|_2 < D, \|\mathbf{b}^{(\tau')}\|_2 < D_b$.

## 5 Experiments

We run two sets of experiments comparing ODEtoODE with several other methods in the supervised setting and to train RL policies with ES. To the best of our knowledge, we are the first to propose Neural ODE architectures for RL-policies and explore how the compactification of the number of parameters they provide can be leveraged by ES methods that benefit from compact models [15].

### 5.1 Neural ODE policies with ODEtoODE architectures

#### 5.1.1 Basic setup

In all Neural ODE methods we were integrating on the time interval $[0, T]$ for $T = 1$ and applied discretization with integration step size $\eta = 0.04$ (in our ODEtoODE we used that $\eta$ for both: the main flow and the orthogonal flow on $\mathcal{O}(d)$). The dimensionality of the embedding of the input state $s$ was chosen to be $h = 64$ for OpenAI Gym Humanoid (for all methods but HyperNet, where we chose $h = 16$, see: discussion below) and $h = 16$ for all other tasks.

Neural ODE policies were obtained by a linear projection of the input state into embedded space and Neural ODE flow in that space, followed by another linear projection into action-space. In addition to learning the parameters of the Neural ODEs, we also trained their initial matrices and those linear projections. Purely linear policies were proven to get good rewards on OpenAI Gym environments [38], yet they lead to inefficient policies in practice [14]; thus even those environments benefit from deep nonlinear policies. We enriched our studies with additional environments from Deep Mind Control Suite. We used standard deviation $\text{stddev} = 0.1$ of the Gaussian noise defining ES perturbations, ES-gradient step size $\delta = 0.01$ and function $\sigma(x) = |x|$ as a nonlinear mapping. In all experiments we used $k = 200$ perturbations per iteration [15].

**Number of policy parameters:** To avoid favoring ODEtoODEs, the other architectures were scaled in such a way that they has similar (but not smaller) number of parameters. The ablation studies were conducted for them across different sizes and those providing best (in terms of the final reward) mean curves were chosen. No ablation studies were run on ODEtoODEs.

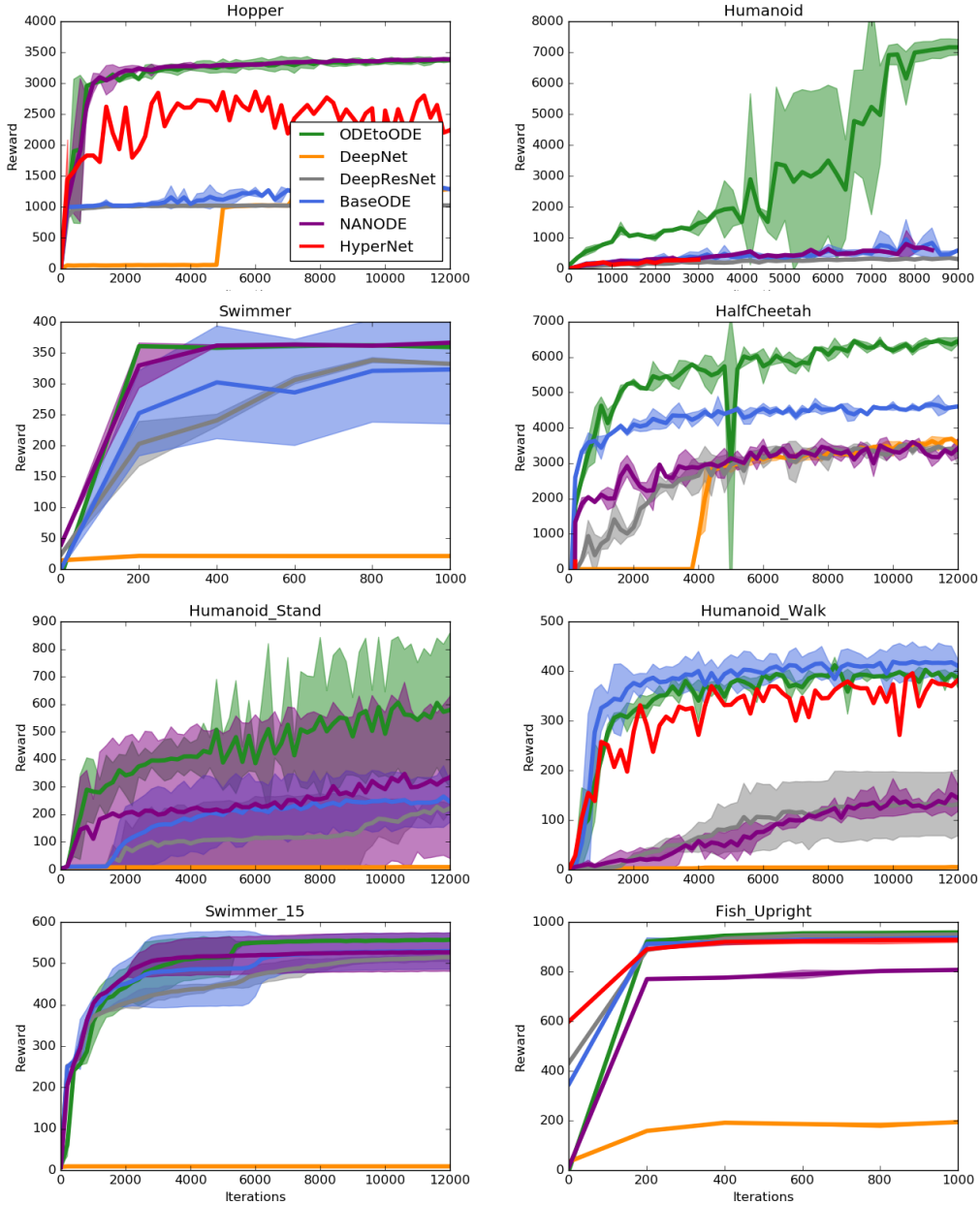

Figure 2: Comparison of different algorithms: ODEtoODE, DeepNet, DeepResNet, BaseODE, NANODE and HyperNet on four OpenAI Gym and four DeepMind Control Suite tasks. HyperNet on Swimmer, Swimmer_15 and Humanoid_Stand as well as DeepNet on Humanoid were not learning at all so corresponding curves were excluded. For HalfCheetah and HyperNet, there was initially a learning signal (red spike at the plot near the origin) but then curve flattened at 0. Each plot shows mean +- stdev across $s = 10$ seeds. HyperNet was also much slower than all other methods because of unstructured hypernetwork computations. Humanoid corresponds to OpenAI Gym environment and two its other versions on the plots to DeepMind Control Suite environment (with two different tasks).

### 5.1.2 Tested methods

We compared the following algorithms including standard deep neural networks and Neural ODEs:

**ODEtoODE:** Matrices corresponding to linear projections were constrained to be taken from Stiefel manifold $\mathcal{ST}(d)$ [14] (a generalization of the orthogonal group $\mathcal{O}(d)$). Evolution strategies (ES) were applied as follows to optimize entire policy. Gradients of Gaussian smoothings of the function: $F : \mathbb{R}^m \to \mathbb{R}$ mapping vectorized policy (we denote as $m$ the number of its parameters) to

obtained reward were computed via standard Monte Carlo procedure [15]. For the part of the gradient vector corresponding to linear projection matrices and initial matrix of the flow, the corresponding Riemannian gradients were computed to make sure that their updates keep them on the respective manifolds [14]. The Riemannian gradients were then used with the exact exponential mapping from $\mathrm{Skew}(d)$ to $\mathcal{O}(d)$. For unconstrained parameters defining orthogonal flow, standard ES-update procedure was used [45]. We applied **ISO-ODEtoODE** version of the method (see: Sec. 3.1).

**Deep(Res)Net:** In this case we tested unstructured deep feedforward fully connected (ResNet) neural network policy with $t = 25$ hidden layers.

**BaseODE:** Standard Neural ODE $\frac{d\mathbf{x}(t)}{dt} = f(\mathbf{x}_t)$, where $f$ was chosen to be a feedforward fully connected network with with two hidden layers of size $h$.

**NANODE:** This leverages recently introduced class of Non-Autonomous Neural ODEs (NAN-ODEs) [19] that were showed to substantially outperform regular Neural ODEs in supervised training [19]. NANODEs rely on flows of the form: $\frac{d\mathbf{x}(t)}{dt} = \sigma(\mathbf{W}_t\mathbf{x}_t)$. Entries of weight matrices $\mathbf{W}_t$ are values of $d$-degree trigonometric polynomials [19] with learnable coefficients. In all experiments we used $d = 5$. We observed that was an optimal choice and larger values of $d$, even though improving model capacity, hurt training when number of perturbations was fixed (as mentioned above, in all experiments we used $k = 200$).

**HyperNet:** For that method [55], matrix $\mathbf{W}_t$ was obtained as a result of the entangled neural ODE in time $t$ after its output de-vectorization (to be more precise: the output of size 256 was reshaped into a matrix from $\mathbb{R}^{16 \times 16}$). We encoded the dynamics of that entangled Neural ODE by a neural network $f$ with two hidden layers of size $s = 16$ each. Note that even relatively small hypernetworks are characterized by large number of parameters which becomes a problem while training ES policies. We thus used $h = 16$ for all tasks while running HyperNet algorithm. We did not put any structural assumptions on matrices $\mathbf{W}_t$, in particular they were not constrained to belong to $\mathcal{O}(d)$.

### 5.1.3 Discussion of the results

The results are presented in Fig. 2. Our ODEtoODE is solely the best performing algorithm on four out of eight tasks: Humanoid, HalfCheetah, Humanoid_Stand and Swimmer_15 and is one of the two best performing on the remaining four. It is clearly the most consistent algorithm across the board and the only one that learns good policies for Humanoid. Each of the remaining methods fails on at least one of the tasks, some such as: HyperNet, DeepNet and DepResNet on more. Exponential mapping (Sec. 3.2) computations for ODEtoODEs with hidden representations $h \le 64$ took negligible time (we used well-optimized scipy.linalg.expm function). Thus all the algorithms but HyperNet (with expensive hypernetwork computations) had similar running time.

## 5.2 Supervised learning with ODEtoODE architectures

We also show that ODEtoODE can be effectively applied in supervised learning by comparing it with multiple baselines on various image datasets.

### 5.2.1 Basic setup

All our supervised learning experiments use the Corrupted MNIST [41] (11 different variants) dataset. For all models in Table 1, we did not use dropout, applied hidden width $w = 128$, and trained for 100 epochs. For all models in Table 2, we used dropout with $r = 0.1$ rate, hidden width $w = 256$, and trained for 100 epochs. For ODEtoODE variants, we used a discretization of $\eta = 0.01$.

### 5.2.2 Tested methods

We compared our ODEtoODE approach with several strong baselines: feedforward fully connected neural networks (Dense), Neural ODEs inspired by [13] (NODE), Non-Autonomous Neural ODEs [19] (NANODE) and hypernets [55] (HyperNet). For NANODE, we vary the degree of the trigonometric polynomials. For HyperNet, we used its gated version and compared against **gated-ODEEtoODE** (see: Sec. 3.1). In this experiment, we focus on fully-connected architecture variants of all models. As discussed in prior work, orthogonality takes on a different form in the convolutional case [49], so we reserve discussion of this framing for future work.

### 5.2.3 Discussion of the results

The results are presented in Table 1 and Table 2 below. Our ODEtoODE outperforms other model variants on 11 out of the 15 tasks in Table 1. On this particularly task, we find that ODEtoODE-1,

whereby we apply a constant perturbation to the hidden units $\theta(0)$ works best. We highlighted the best results for each task in **bolded blue**.

| Models | Dense-1 | Dense-10 | NODE | NANODE-1 | NANODE-10 | HyperNet-1 | HyperNet-10 | ODEtoODE-1 | ODEtoODE-10 |
|---|---|---|---|---|---|---|---|---|---|
| **Dotted Lines** | 92.99 | 88.22 | 91.54 | 92.42 | 92.74 | 91.88 | 91.9 | **95.42** | 95.22 |
| **Spatter** | 93.54 | 89.52 | 92.49 | 93.13 | 93.15 | 93.19 | 93.28 | **94.9** | **94.9** |
| **Stripe** | 30.55 | 20.57 | 36.76 | 16.4 | 21.37 | 19.71 | 18.69 | **44.51** | 44.37 |
| **Translate** | 25.8 | 24.82 | 27.09 | 28.97 | 29.31 | **29.42** | 29.3 | 26.82 | 26.63 |
| **Rotate** | 82.8 | 80.38 | 82.76 | 83.19 | 83.65 | 83.5 | 83.56 | 83.9 | **84.1** |
| **Scale** | 58.68 | 62.73 | 62.05 | 66.43 | 66.63 | 67.84 | **68.11** | 66.68 | 66.76 |
| **Shear** | 91.73 | 89.52 | 91.82 | 92.18 | 93.11 | 92.33 | 92.48 | **93.37** | 92.93 |
| **Motion Blur** | 78.16 | 67.25 | 75.18 | 76.53 | **79.22** | 78.82 | 78.33 | 78.63 | 77.58 |
| **Glass Blur** | 91.62 | 84.89 | 87.94 | 90.18 | 91.07 | 91.3 | 91.17 | **93.91** | 93.29 |
| **Shot Noise** | 96.16 | 91.63 | 94.24 | 94.97 | 94.73 | 94.76 | 94.81 | **96.91** | 96.71 |
| **Identity** | 97.55 | 95.73 | 97.61 | 97.65 | 97.69 | 97.72 | 97.54 | **97.94** | 97.88 |

Table 1: Test accuracy comparison of different methods. Postfix terms refer to hidden depth for Dense, trigonometric polynomial degree for NANODE, and number of gates for HyperNet and ODEtoODE.

| Models | Dense-1 | Dense-2 | Dense-4 | NODE | NANODE-1 | NANODE-2 | NANODE-4 | HyperNet-1 | HyperNet-2 | HyperNet-4 | ODEtoODE-1 | ODEtoODE-2 | ODEtoODE-4 |
|---|---|---|---|---|---|---|---|---|---|---|---|---|---|
| **Dotted Line** | 94.47 | 93.21 | 91.88 | 92.9 | 92 | 92.02 | 92.02 | 92.35 | 92.91 | 92.57 | 95.64 | **95.66** | 95.6 |
| **Spatter** | 94.52 | 93.59 | 93.63 | 93.32 | 92.81 | 92.82 | 92.84 | 94.09 | 93.94 | 93.73 | 95.28 | **95.47** | 95.29 |
| **Stripe** | 29.69 | 32.73 | 31.49 | 36.08 | 27.32 | 23.56 | 24.66 | 30.86 | 31.12 | 29.1 | **36.25** | 28.21 | 31.91 |
| **Translate** | 25.85 | 28.35 | 27.27 | 29.13 | 29.11 | 29.24 | 28.7 | 29.85 | 29.68 | **29.87** | 25.61 | 26.1 | 26.42 |
| **Rotate** | 83.62 | 83.73 | 83.88 | 83.09 | 82.5 | 82.77 | 83.03 | 83.96 | 84.04 | 84.13 | **85.1** | 85.03 | 84.72 |
| **Scale** | 61.41 | 65.07 | 63.72 | 63.62 | 65.49 | 65.33 | 64.03 | 69.51 | 68.77 | **69.8** | 67.97 | 66.95 | 67.09 |
| **Shear** | 92.25 | 92.76 | 92.55 | 92.27 | 92.08 | 92.18 | 92.3 | 93.35 | 92.84 | 93.04 | 93.15 | 93.06 | **93.38** |
| **Motion Blur** | 76.47 | 73.89 | 74.95 | 76.3 | 75.24 | 75.88 | 76.22 | 80.67 | 81.26 | **81.36** | 78.92 | 79.11 | 79.02 |
| **Glass Blur** | 92.5 | 89.65 | 90.29 | 89.47 | 89.5 | 89.5 | 89.8 | 93.07 | 92.67 | 92.69 | **94.46** | 94.19 | 94.3 |
| **Shot Noise** | 96.41 | 95.78 | 95.22 | 95.09 | 94.18 | 94.12 | 93.85 | 95.36 | 96.88 | 96.77 | **96.93** | 96.88 | 96.77 |
| **Identity** | 97.7 | 97.75 | 97.71 | 97.64 | 97.6 | 97.5 | 97.52 | 97.7 | 97.82 | 97.79 | 98.03 | **98.12** | 98.11 |

Table 2: Additional Sweep for the setting as in Table 1. This time all models also incorporate dropout with rate r = 0.1 and width w = 256. As above in Table 1, ODEtoODE variants achieve the highest performance.

# 6   Conclusion

In this paper, we introduced nested Neural ODE systems, where the parameter-flow evolves on the orthogonal group $\mathcal{O}(d)$. Constraining this matrix-flow to develop on the compact manifold provides us with an architecture that can be efficiently trained without exploding/vanishing gradients, as we showed theoretically and demonstrated empirically by presenting gains on various downstream tasks. We are the first to demonstrate that algorithms training RL policies and relying on compact models' representations can significantly benefit from such hierarchical systems.

# 7   Broader impact

We believe our contributions have potential broader impact that we briefly discuss below:

**Reinforcement Learning with Neural ODEs:**   To the best of our knowledge, we are the first to propose to apply nested Neural ODEs in Reinforcement Learning, in particular to train Neural ODE policies. More compact architectures encoding deep neural network systems is an especially compelling feature in policy training algorithms, in particular while combined with ES methods admitting embarrassingly simple and efficient parallelization, yet suffering from high sampling complexity that increases with the number of policy parameters. Our work shows that RL training of such systems can be conducted efficiently provided that evolution of the parameters of the system is highly structured and takes place on compact matrix manifolds.

**Learnable Isospectral Flows:**   We demonstrated that ISO-ODEtoODEs can be successfully applied to learn reinforcement learning policies. Those rely on the isospectral flows that in the past were demonstrated to be capable of solving combinatorial problems ranging from sorting to (graph) matching [8, 54]. As emphasized before, such flows are however fixed and not trainable whereas we learn ours. That suggests that isospectral flows can be potentially learned to solve combinatorially-flavored machine learning problems or even integrated with non-combinatorial blocks in larger ML computational pipelines. The benefits of such an approach lie in the fact that we can efficiently backpropagate through these continuous systems and is related to recent research on differentiable sorting [18, 5, 24].

# 8   Acknowledgements

Adrian Weller acknowledges support from the David MacKay Newton research fellowship at Darwin College, The Alan Turing Institute under EPSRC grant EP/N510129/1 and U/B/000074, and the Leverhulme Trust via CFI. Valerii Likhosherstov acknowledges support from the Cambridge Trust and DeepMind.

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
