[Supplementary Material]

## APPENDIX: An Ode to an ODE

## 9 Proof of Lemma 4.1

*Proof.* Consider the following Euler-based discretization of the main (non-matrix) flow of the ODEtoODE:

$$\mathbf{x}_{\frac{i+1}{N}} = \mathbf{x}_{\frac{i}{N}} + \frac{1}{N}f\left(\mathbf{W}^{\theta}\left(\frac{i}{N}\right)\mathbf{x}_{\frac{i}{N}}\right), \tag{10}$$

where $i = 0, 1, ., , , N - 1$ and $N \in \mathbb{N}_+$ is fixed (defines granularity of discretization). Denote: $\mathbf{a}_i^N = \mathbf{x}_{\frac{i}{N}}$, $\mathbf{c}_i^N = \mathbf{b}_{\frac{i}{N}}$, $\mathbf{V}_i^N = \mathbf{W}^{\theta}(\frac{i}{N})$ and $\mathbf{z}_{i+1}^N = \mathbf{V}_{i+1}^{N,\theta}\mathbf{a}_i^N + \mathbf{c}_{i+1}^N$. We obtain the following discrete dynamical system:

$$\mathbf{a}_{i+1}^N = \mathbf{a}_i^N + \frac{1}{N}f\left(\mathbf{z}_{i+1}^N\right), \tag{11}$$

for $i = 0, 1, ..., N - 1$.

Let $\mathcal{L} = \mathcal{L}(\mathbf{x}_1) = \mathcal{L}(\mathbf{a}_N^N)$ by the loss function. Our first goal is to compute $\frac{\partial \mathcal{L}}{\partial \mathbf{a}_i^N}$ for $i = 0, ..., N - 1$.

Using Equation 11, we get:

$$\frac{\partial \mathcal{L}}{\partial \mathbf{a}_i^N} = \frac{\partial \mathbf{a}_{i+1}^N}{\partial \mathbf{a}_i^N}\frac{\partial \mathcal{L}}{\partial \mathbf{a}_{i+1}^N} = (\mathbf{I}_d + \frac{1}{N}\text{diag}(f'(\mathbf{z}_{i+1}^N))\frac{\partial \mathbf{z}_{i+1}^N}{\partial \mathbf{a}_i^N})\frac{\partial \mathcal{L}}{\partial \mathbf{a}_{i+1}^N} =$$
$$(\mathbf{I}_d + \frac{1}{N}\text{diag}(f'(\mathbf{z}_{i+1}^N))\mathbf{V}_{i+1}^{N,\theta})\frac{\partial \mathcal{L}}{\partial \mathbf{a}_{i+1}^N} \tag{12}$$

Therefore we conclude that:

$$\frac{\partial \mathcal{L}}{\partial \mathbf{a}_i^N} = \left[\prod_{r=i+1}^{N}(\mathbf{I}_d + \frac{1}{N}\text{diag}(f'(\mathbf{z}_r^N))\mathbf{V}_r^{N,\theta})\right]\frac{\partial \mathcal{L}}{\partial \mathbf{a}_N^N} \tag{13}$$

Note that for function $f$ such that $|f'(x)| = 1$ in its differentiable points we have: $\text{diag}(f'(\mathbf{z}_r^N))$ is a diagonal matrix with nonzero entries taken from $\{-1, +1\}$, in particular $D \in \mathcal{O}(d)$, where $\mathcal{O}(d)$ stands for the *orthogonal group*.

Define: $\mathbf{G}_r^N = \text{diag}(f'(\mathbf{z}_l^N))\mathbf{V}_r^{N,\theta}$. Thus we have:

$$\frac{\partial \mathcal{L}}{\partial \mathbf{a}_i^N} = \left[\prod_{r=i+1}^{N}(\mathbf{I}_d + \frac{1}{N}\mathbf{G}_r^N)\right]\frac{\partial \mathcal{L}}{\partial \mathbf{a}_N^N} \tag{14}$$

Note that the following is true:

$$\prod_{r=i+1}^{N}(\mathbf{I}_d + \frac{1}{N}\mathbf{G}_r^N) = \sum_{\{r_1,...,r_k\}\subseteq\{i+1,...,N\}}\frac{1}{N^k}\mathbf{G}_{r_1}^N \cdot ... \cdot \mathbf{G}_{r_k}^N \tag{15}$$

Therefore we can conclude that:

$$\|\prod_{r=i+1}^{N}(\mathbf{I}_d + \frac{1}{N}\mathbf{G}_r^N)\|_2 \le \sum_{\{r_1,...,r_k\}\subseteq\{i+1,...,N\}}\frac{1}{N^k}\|\mathbf{G}_{r_1}^N \cdot ... \cdot \mathbf{G}_{r_k}^N\|_2$$
$$= \sum_{k=0}^{N-i}\frac{1}{N^k}\binom{N-i}{k} = \sum_{k=0}^{N-i}\frac{(N-i-k+1)\cdot...\cdot(N-i)}{N^k k!} \le \sum_{k=0}^{N-i}\frac{1}{k!} \le e, \tag{16}$$

where we used the fact that $\mathcal{O}(d)$ is a group under matrix-multiplication and $\|\mathbf{G}\|_2 = 1$ for every $\mathbf{G} \in \mathcal{O}(d)$. That proves inequality: $\|\frac{\partial \mathcal{L}}{\partial \mathbf{a}_i^N}\|_2 \le e\|\frac{\partial \mathcal{L}}{\partial \mathbf{a}_N^N}\|_2$.

To prove the other inequality: $\|\frac{\partial \mathcal{L}}{\partial \mathbf{a}_i^N}\|_2 \ge (\frac{1}{e} - \epsilon)\|\frac{\partial \mathcal{L}}{\partial \mathbf{a}_N^N}\|_2$, for large enough $N \ge N(\epsilon)$, it suffices to observe that if $\mathbf{G} \in \mathcal{O}(d)$, then for any $\mathbf{v} \in \mathbb{R}^d$ we have (by triangle inequality):

$$\|(\mathbf{I}_d + \frac{1}{N}\mathbf{G})\mathbf{v}\|_2 \ge \|\mathbf{v}\|_2 - \|\frac{1}{N}\mathbf{G}\mathbf{v}\|_2 = (1 - \frac{1}{N})\|\mathbf{v}\|_2. \tag{17}$$

Lemma 4.1 then follows immediately from the fact that sequence: $\{a_N : N = 1, 2, 3, ...\}$ defined as: $a_N = (1 - \frac{1}{N})^N$ has limit $e^{-1}$ and by taking $N \to \infty$. □

## 10 Proof of Theorem 1

We first prove a number of useful Lemmas. In our derivations we frequently employ an inequality stating that for $\alpha > 0$ $(1 + \frac{\alpha}{N})^N = \exp(N \log(1 + \frac{\alpha}{N})) \leq \exp(N \cdot \frac{\alpha}{N}) = e^\alpha$ which follows from $\exp(\cdot)$'s monotonicity and $\log(\cdot)$'s concavity and that $\log(1) = 0, \log'(1) = 1$.

**Lemma 1.** *If Assumption 4.3 is satisfied, then for any $\theta' = \{\Omega_1', \Omega_2', \mathbf{b}', \mathbf{N}', \mathbf{Q}', \mathbf{W}_0'\} \in \mathbb{D}$ and $\theta'' = \{\Omega_1'', \Omega_2'', \mathbf{b}'', \mathbf{N}'', \mathbf{Q}'', \mathbf{W}_0''\} \in \mathbb{D}$ such that $\|\mathbf{N}'\|_2, \|\mathbf{Q}'\|_2, \|\mathbf{N}''\|_2, \|\mathbf{Q}''\|_2 \leq D, \|\mathbf{b}'\|_2, \|\mathbf{b}''\|_2 \leq D_b$ for some $D, D_b > 0$ it holds that*

$$\forall \mathbf{s}', \mathbf{s}'' \in \mathbb{R}^d : \|g_{\theta'}(\mathbf{s}') - g_{\theta''}(\mathbf{s}'')\|_2 \leq e\|\mathbf{s}' - \mathbf{s}''\|_2$$

$$+ \left(e\|\mathbf{s}''\|_2 + (e-1)D_b\right)\left(1 + (e-1)((1 + \frac{1}{D})e^{4D^2} - \frac{1}{D}))\right)\|\theta' - \theta''\|_2, \tag{18}$$

$$\|g_{\theta''}(\mathbf{s}'')\|_2 \leq e\|\mathbf{s}''\|_2 + (e-1)D_b. \tag{19}$$

*Proof.* Indeed,

$$\|g_{\theta'}(\mathbf{s}') - g_{\theta''}(\mathbf{s}'')\|_2 = \|g_{\theta'}(\mathbf{s}') - g_{\theta'}(\mathbf{s}'') + g_{\theta'}(\mathbf{s}'') - g_{\theta''}(\mathbf{s}'')\|_2$$

$$\leq \|g_{\theta'}(\mathbf{s}') - g_{\theta'}(\mathbf{s}'')\|_2 + \|g_{\theta'}(\mathbf{s}'') - g_{\theta''}(\mathbf{s}'')\|_2. \tag{20}$$

Let $\mathbf{x}_1', \ldots, \mathbf{x}_N'$ and $\mathbf{x}_1'', \ldots, \mathbf{x}_N''$ be rollouts (5-6) corresponding to computation of $g_{\theta'}(s')$ and $g_{\theta'}(s'')$ respectively. For any Stiefel matrix $\Omega \in \mathcal{ST}(d_1, d_2)$ (including square orthogonal matrices) it holds that $\|\Omega\|_2 = 1$. We use it to deduce:

$$\|g_{\theta'}(\mathbf{s}') - g_{\theta'}(\mathbf{s}'')\|_2 = \|\Omega_2'\mathbf{x}_N' - \Omega_2'\mathbf{x}_N''\|_2 \leq \|\Omega_2'\|_2\|\mathbf{x}_N' - \mathbf{x}_N''\|_2 = \|\mathbf{x}_N' - \mathbf{x}_N''\|_2$$

$$= \|\mathbf{x}_{N-1}' - \mathbf{x}_{N-1}'' + \frac{1}{N}\left(f(\mathbf{W}_N\mathbf{x}_{N-1}' + \mathbf{b}) - f(\mathbf{W}_N\mathbf{x}_{N-1}'' + \mathbf{b})\right)\|_2$$

$$\leq \|\mathbf{x}_{N-1}' - \mathbf{x}_{N-1}''\|_2 + \frac{1}{N}\|f(\mathbf{W}_N\mathbf{x}_{N-1}' + \mathbf{b}) - f(\mathbf{W}_N\mathbf{x}_{N-1}'' + \mathbf{b})\|_2$$

$$\leq \|\mathbf{x}_{N-1}' - \mathbf{x}_{N-1}''\|_2 + \frac{1}{N}\|\mathbf{W}_N\mathbf{x}_{N-1}' - \mathbf{W}_N\mathbf{x}_{N-1}''\|_2$$

$$= \|\mathbf{x}_{N-1}' - \mathbf{x}_{N-1}''\|_2 + \frac{1}{N}\|\mathbf{x}_{N-1}' - \mathbf{x}_{N-1}''\|_2$$

$$= (1 + \frac{1}{N})\|\mathbf{x}_{N-1}' - \mathbf{x}_{N-1}''\|_2 \leq \cdots \leq (1 + \frac{1}{N})^N\|\mathbf{x}_0' - \mathbf{x}_0''\|_2$$

$$\leq e\|\mathbf{x}_0' - \mathbf{x}_0''\|_2^2 = e\|\Omega_1'\mathbf{s}' - \Omega_1'\mathbf{s}''\|_2^2 \leq e\|\Omega_1'\|_2\|\mathbf{s}' - \mathbf{s}''\|_2^2 = e\|\mathbf{s}' - \mathbf{s}''\|_2^2. \tag{21}$$

Let $\mathbf{x}_1', \mathbf{W}_1', \ldots, \mathbf{x}_N', \mathbf{W}_N'$ and $\mathbf{x}_1'', \mathbf{W}_1'', \ldots, \mathbf{x}_N'', \mathbf{W}_N''$ be rollouts (5-6) corresponding to computation of $g_{\theta'}(s'')$ and $g_{\theta''}(s'')$ respectively. We fix $i \in \{1, \ldots, N\}$ and deduce, using Assumption 4.3 in particular, that

$$\|\mathbf{x}_i''\|_2 \leq \|\mathbf{x}_{i-1}''\|_2 + \frac{1}{N}\|f(\mathbf{W}_i''\mathbf{x}_{i-1}'' + \mathbf{b}'')\|_2 \leq \|\mathbf{x}_{i-1}''\|_2 + \frac{1}{N}\|\mathbf{W}_i''\mathbf{x}_{i-1}'' + \mathbf{b}''\|_2$$

$$\leq \|\mathbf{x}_{i-1}''\|_2 + \frac{1}{N}\|\mathbf{W}_i''\mathbf{x}_{i-1}''\|_2 + \frac{1}{N}\|\mathbf{b}''\|_2 = (1 + \frac{1}{N})\|\mathbf{x}_{i-1}''\|_2 + \frac{1}{N}\|\mathbf{b}''\|_2$$

$$\leq (1 + \frac{1}{N})^i\|\mathbf{x}_0''\|_2 + \frac{1}{N}\|\mathbf{b}''\|_2\sum_{j=0}^{i-1}(1 + \frac{1}{N})^j = (1 + \frac{1}{N})^i\|\Omega_1''\mathbf{s}''\|_2$$

$$+ ((1 + \frac{1}{N})^i - 1)\|\mathbf{b}''\|_2 = (1 + \frac{1}{N})^i\|\Omega_1''\|_2\|\mathbf{s}''\|_2 + ((1 + \frac{1}{N})^i - 1)D_b$$

$$\leq (1 + \frac{1}{N})^N\|\mathbf{s}''\|_2 + ((1 + \frac{1}{N})^N - 1)D_b$$

$$\leq e\|\mathbf{s}''\|_2 + (e-1)D_b. \tag{22}$$

As a particular case of (22) when $i = N$ and $\|g_{\theta''}(\mathbf{s}'')\|_2 = \|\mathbf{\Omega}_2'' \mathbf{x}_N''\|_2 \leq \|\mathbf{x}_N''\|_2$ we obtain (19). Set

$$\mathbf{A}' = \frac{1}{N}(\mathbf{W}_{i-1}'^\top \mathbf{Q}' \mathbf{W}_{i-1}' \mathbf{N}' - \mathbf{N}' \mathbf{W}_{i-1}'^\top \mathbf{Q}' \mathbf{W}_{i-1}'),$$

$$\mathbf{A}'' = \frac{1}{N}(\mathbf{W}_{i-1}''^\top \mathbf{Q}'' \mathbf{W}_{i-1}'' \mathbf{N}'' - \mathbf{N}'' \mathbf{W}_{i-1}''^\top \mathbf{Q}'' \mathbf{W}_{i-1}'').$$

Since $\mathbf{A}', \mathbf{A}'' \in \mathrm{Skew}(d)$ and $\mathrm{Skew}(d)$ is a vector space, we conclude that $\exp(\alpha \mathbf{A}' + \alpha t(\mathbf{A}'' - \mathbf{A}')) \in \mathcal{O}(d)$ for any $t, \alpha \in \mathbb{R}$ where we use that $\exp(\cdot)$ maps $\mathrm{Skew}(d)$ into $\mathcal{O}(d)$. We also use a rule [51] which states that for $\mathbf{X} : \mathbb{R} \rightarrow \mathbb{R}^{d \times d}$

$$\frac{d}{dt} \exp(\mathbf{X}(t)) = \int_0^1 \exp(\alpha \mathbf{X}(t)) \frac{d\mathbf{X}(t)}{dt} \exp((1 - \alpha)\mathbf{X}(t)) d\alpha$$

to deduce that

$$\| \exp(\mathbf{A}') - \exp(\mathbf{A}'') \|_2^2 = \| \int_{t=0}^1 \frac{d}{dt} \exp(\mathbf{A}'' + t(\mathbf{A}' - \mathbf{A}'')) dt \|_2^2$$

$$= \| \int_0^1 \int_0^1 \exp\Big( \alpha \mathbf{A}'' + \alpha t(\mathbf{A}' - \mathbf{A}'') \Big)(\mathbf{A}' - \mathbf{A}'') \exp\Big( (1 - \alpha)\mathbf{A}''$$

$$+ (1 - \alpha)t(\mathbf{A}' - \mathbf{A}'') \Big) d\alpha dt \|_2^2$$

$$\leq \int_0^1 \int_0^1 \| \exp\Big( \alpha \mathbf{A}'' + \alpha t(\mathbf{A}' - \mathbf{A}'') \Big)(\mathbf{A}' - \mathbf{A}'') \exp\Big( (1 - \alpha)\mathbf{A}''$$

$$+ (1 - \alpha)t(\mathbf{A}' - \mathbf{A}'') \Big) \|_2^2 d\alpha dt$$

$$= \int_0^1 \int_0^1 \| \mathbf{A}' - \mathbf{A}'' \|_2^2 d\alpha dt = \| \mathbf{A}' - \mathbf{A}'' \|_2^2.$$

Consequently, we derive that

$\|\mathbf{W}_i' - \mathbf{W}_i''\|_2 = \|\mathbf{W}_{i-1}' \exp(\mathbf{A}') - \mathbf{W}_{i-1}'' \exp(\mathbf{A}'')\|_2$

$= \|\mathbf{W}_{i-1}' \exp(\mathbf{A}') - \mathbf{W}_{i-1}'' \exp(\mathbf{A}') + \mathbf{W}_{i-1}'' \exp(\mathbf{A}') - \mathbf{W}_{i-1}'' \exp(\mathbf{A}'')\|_2$

$\leq \|\mathbf{W}_{i-1}' \exp(\mathbf{A}') - \mathbf{W}_{i-1}'' \exp(\mathbf{A}')\|_2 + \|\mathbf{W}_{i-1}'' \exp(\mathbf{A}') - \mathbf{W}_{i-1}'' \exp(\mathbf{A}'')\|_2$

$= \|\mathbf{W}_{i-1}' - \mathbf{W}_{i-1}''\|_2 + \| \exp(\mathbf{A}') - \exp(\mathbf{A}'')\|_2 \leq \|\mathbf{W}_{i-1}' - \mathbf{W}_{i-1}''\|_2 + \|\mathbf{A}' - \mathbf{A}''\|_2$

$= \|\mathbf{W}_{i-1}' - \mathbf{W}_{i-1}''\|_2 + \frac{1}{N}\|(\mathbf{W}_{i-1}'^\top \mathbf{Q}' \mathbf{W}_{i-1}' \mathbf{N}' - \mathbf{W}_{i-1}''^\top \mathbf{Q}'' \mathbf{W}_{i-1}'' \mathbf{N}'') - (\mathbf{N}' \mathbf{W}_{i-1}'^\top \mathbf{Q}' \mathbf{W}_{i-1}'$

$- \mathbf{N}'' \mathbf{W}_{i-1}''^\top \mathbf{Q}'' \mathbf{W}_{i-1}'')\|_2$

$\leq \|\mathbf{W}_{i-1}' - \mathbf{W}_{i-1}''\|_2 + \frac{1}{N}\|\mathbf{W}_{i-1}'^\top \mathbf{Q}' \mathbf{W}_{i-1}' \mathbf{N}' - \mathbf{W}_{i-1}''^\top \mathbf{Q}'' \mathbf{W}_{i-1}'' \mathbf{N}''\|_2 + \frac{1}{N}\|\mathbf{N}' \mathbf{W}_{i-1}'^\top \mathbf{Q}' \mathbf{W}_{i-1}'$

$- \mathbf{N}'' \mathbf{W}_{i-1}''^\top \mathbf{Q}'' \mathbf{W}_{i-1}''\|_2$

$= \|\mathbf{W}_{i-1}' - \mathbf{W}_{i-1}''\|_2 + \frac{1}{N}\|\mathbf{W}_{i-1}'^\top \mathbf{Q}' \mathbf{W}_{i-1}' \mathbf{N}' - \mathbf{W}_{i-1}'^\top \mathbf{Q}' \mathbf{W}_{i-1}'' \mathbf{N}'' + \mathbf{W}_{i-1}'^\top \mathbf{Q}' \mathbf{W}_{i-1}'' \mathbf{N}''$

$- \mathbf{W}_{i-1}''^\top \mathbf{Q}'' \mathbf{W}_{i-1}'' \mathbf{N}''\|_2 + \frac{1}{N}\|\mathbf{N}' \mathbf{W}_{i-1}'^\top \mathbf{Q}' \mathbf{W}_{i-1}' - \mathbf{N}' \mathbf{W}_{i-1}'^\top \mathbf{Q}'' \mathbf{W}_{i-1}'' + \mathbf{N}' \mathbf{W}_{i-1}'^\top \mathbf{Q}'' \mathbf{W}_{i-1}''$

$- \mathbf{N}'' \mathbf{W}_{i-1}''^\top \mathbf{Q}'' \mathbf{W}_{i-1}''\|_2$

$\leq \|\mathbf{W}_{i-1}' - \mathbf{W}_{i-1}''\|_2 + \frac{1}{N}\|\mathbf{W}_{i-1}'^\top \mathbf{Q}' \mathbf{W}_{i-1}' \mathbf{N}' - \mathbf{W}_{i-1}'^\top \mathbf{Q}' \mathbf{W}_{i-1}'' \mathbf{N}''\|_2 + \frac{1}{N}\|\mathbf{W}_{i-1}'^\top \mathbf{Q}' \mathbf{W}_{i-1}'' \mathbf{N}''$

$- \mathbf{W}_{i-1}''^\top \mathbf{Q}'' \mathbf{W}_{i-1}'' \mathbf{N}''\|_2 + \frac{1}{N}\|\mathbf{N}' \mathbf{W}_{i-1}'^\top \mathbf{Q}' \mathbf{W}_{i-1}' - \mathbf{N}' \mathbf{W}_{i-1}'^\top \mathbf{Q}'' \mathbf{W}_{i-1}''\|_2 + \frac{1}{N}\|\mathbf{N}' \mathbf{W}_{i-1}'^\top \mathbf{Q}'' \mathbf{W}_{i-1}''$

$- \mathbf{N}'' \mathbf{W}_{i-1}''^\top \mathbf{Q}'' \mathbf{W}_{i-1}''\|_2$

$\leq \|\mathbf{W}_{i-1}' - \mathbf{W}_{i-1}''\|_2 + \frac{1}{N}\|\mathbf{Q}'\|_2 \|\mathbf{W}_{i-1}' \mathbf{N}' - \mathbf{W}_{i-1}'' \mathbf{N}''\|_2 + \frac{1}{N}\|\mathbf{N}''\|_2 \|\mathbf{W}_{i-1}'^\top \mathbf{Q}' - \mathbf{W}_{i-1}''^\top \mathbf{Q}''\|_2$

$$+ \frac{1}{N}\|\mathbf{N}'\|_2\|\mathbf{Q}'\mathbf{W}'_{i-1} - \mathbf{Q}''\mathbf{W}''_{i-1}\|_2 + \frac{1}{N}\|\mathbf{Q}''\|_2\|\mathbf{N}'\mathbf{W}'^{\top}_{i-1} - \mathbf{N}''\mathbf{W}''^{\top}_{i-1}\|_2$$

$$\leq \|\mathbf{W}'_{i-1} - \mathbf{W}''_{i-1}\|_2 + \frac{D}{N}\|\mathbf{W}'_{i-1}\mathbf{N}' - \mathbf{W}''_{i-1}\mathbf{N}''\|_2 + \frac{D}{N}\|\mathbf{W}'^{\top}_{i-1}\mathbf{Q}' - \mathbf{W}''^{\top}_{i-1}\mathbf{Q}''\|_2$$

$$+ \frac{D}{N}\|\mathbf{Q}'\mathbf{W}'_{i-1} - \mathbf{Q}''\mathbf{W}''_{i-1}\|_2 + \frac{D}{N}\|\mathbf{N}'\mathbf{W}'^{\top}_{i-1} - \mathbf{N}''\mathbf{W}''^{\top}_{i-1}\|_2$$

$$= \|\mathbf{W}'_{i-1} - \mathbf{W}''_{i-1}\|_2 + \frac{D}{N}\|\mathbf{W}'_{i-1}\mathbf{N}' - \mathbf{W}'_{i-1}\mathbf{N}'' + \mathbf{W}'_{i-1}\mathbf{N}'' - \mathbf{W}''_{i-1}\mathbf{N}''\|_2 + \frac{D}{N}\|\mathbf{W}'^{\top}_{i-1}\mathbf{Q}'$$

$$- \mathbf{W}'^{\top}_{i-1}\mathbf{Q}'' + \mathbf{W}'^{\top}_{i-1}\mathbf{Q}'' - \mathbf{W}''^{\top}_{i-1}\mathbf{Q}''\|_2 + \frac{D}{N}\|\mathbf{Q}'\mathbf{W}'_{i-1} - \mathbf{Q}'\mathbf{W}''_{i-1} + \mathbf{Q}'\mathbf{W}''_{i-1} - \mathbf{Q}''\mathbf{W}''_{i-1}\|_2$$

$$+ \frac{D}{N}\|\mathbf{N}'\mathbf{W}'^{\top}_{i-1} - \mathbf{N}'\mathbf{W}''^{\top}_{i-1} + \mathbf{N}'\mathbf{W}''^{\top}_{i-1} - \mathbf{N}''\mathbf{W}''^{\top}_{i-1}\|_2$$

$$\leq \|\mathbf{W}'_{i-1} - \mathbf{W}''_{i-1}\|_2 + \frac{D}{N}\|\mathbf{W}'_{i-1}\mathbf{N}' - \mathbf{W}'_{i-1}\mathbf{N}''\|_2 + \frac{D}{N}\|\mathbf{W}'_{i-1}\mathbf{N}'' - \mathbf{W}''_{i-1}\mathbf{N}''\|_2$$

$$+ \frac{D}{N}\|\mathbf{W}'^{\top}_{i-1}\mathbf{Q}' - \mathbf{W}'^{\top}_{i-1}\mathbf{Q}''\|_2 + \frac{D}{N}\|\mathbf{W}'^{\top}_{i-1}\mathbf{Q}'' - \mathbf{W}''^{\top}_{i-1}\mathbf{Q}''\|_2 + \frac{D}{N}\|\mathbf{Q}'\mathbf{W}'_{i-1} - \mathbf{Q}'\mathbf{W}''_{i-1}\|_2$$

$$+ \frac{D}{N}\|\mathbf{Q}'\mathbf{W}''_{i-1} - \mathbf{Q}''\mathbf{W}''_{i-1}\|_2 + \frac{D}{N}\|\mathbf{N}'\mathbf{W}'^{\top}_{i-1} - \mathbf{N}'\mathbf{W}''^{\top}_{i-1}\|_2 + \frac{D}{N}\|\mathbf{N}'\mathbf{W}''^{\top}_{i-1} - \mathbf{N}''\mathbf{W}''^{\top}_{i-1}\|_2$$

$$\leq \|\mathbf{W}'_{i-1} - \mathbf{W}''_{i-1}\|_2 + \frac{D}{N}\|\mathbf{N}' - \mathbf{N}''\|_2 + \frac{D}{N}\|\mathbf{N}''\|_2\|\mathbf{W}'_{i-1} - \mathbf{W}''_{i-1}\|_2 + \frac{D}{N}\|\mathbf{Q}' - \mathbf{Q}''\|_2$$

$$+ \frac{D}{N}\|\mathbf{Q}''\|_2\|\mathbf{W}'^{\top}_{i-1} - \mathbf{W}''^{\top}_{i-1}\|_2 + \frac{D}{N}\|\mathbf{Q}'\|_2\|\mathbf{W}'_{i-1} - \mathbf{W}''_{i-1}\|_2 + \frac{D}{N}\|\mathbf{Q}' - \mathbf{Q}''\|_2$$

$$+ \frac{D}{N}\|\mathbf{N}'\|_2\|\mathbf{W}'^{\top}_{i-1} - \mathbf{W}''^{\top}_{i-1}\|_2 + \frac{D}{N}\|\mathbf{N}' - \mathbf{N}''\|_2$$

$$\leq (1 + 4\frac{D^2}{N})\|\mathbf{W}'_{i-1} - \mathbf{W}''_{i-1}\|_2 + 2\frac{D}{N}\|\mathbf{N}' - \mathbf{N}''\|_2 + 2\frac{D}{N}\|\mathbf{Q}' - \mathbf{Q}''\|_2$$

$$\leq (1 + 4\frac{D^2}{N})\|\mathbf{W}'_{i-1} - \mathbf{W}''_{i-1}\|_2 + 4\frac{D}{N}\|\theta' - \theta''\|_2 \leq \dots$$

$$\leq (1 + 4\frac{D^2}{N})^i\|\mathbf{W}'_0 - \mathbf{W}''_0\|_2 + 4\frac{D}{N}\sum_{j=0}^{i-1}(1 + 4\frac{D^2}{N})^j\|\theta' - \theta''\|_2$$

$$= (1 + 4\frac{D^2}{N})^i\|\mathbf{W}'_0 - \mathbf{W}''_0\|_2 + \frac{1}{D}((1 + 4\frac{D^2}{N})^i - 1)\|\theta' - \theta''\|_2$$

$$\leq (1 + 4\frac{D^2}{N})^i\|\theta' - \theta''\|_2 + \frac{1}{D}((1 + 4\frac{D^2}{N})^i - 1)\|\theta' - \theta''\|_2$$

$$= \left((1 + \frac{1}{D})(1 + 4\frac{D^2}{N})^i - \frac{1}{D}\right)\|\theta' - \theta''\|_2 \leq \left((1 + \frac{1}{D})(1 + 4\frac{D^2}{N})^N - \frac{1}{D}\right)\|\theta' - \theta''\|_2$$

$$\leq \left((1 + \frac{1}{D})e^{4D^2} - \frac{1}{D}\right)\|\theta' - \theta''\|_2$$

We use (22) and derive that

$$\|\mathbf{x}'_i - \mathbf{x}''_i\|_2 = \|\mathbf{x}'_{i-1} - \mathbf{x}''_{i-1} + \frac{1}{N}(f(\mathbf{W}'_i\mathbf{x}'_{i-1} + \mathbf{b}) - f(\mathbf{W}''_i\mathbf{x}''_{i-1} + \mathbf{b}))\|_2$$

$$\leq \|\mathbf{x}'_{i-1} - \mathbf{x}''_{i-1}\|_2 + \frac{1}{N}\|f(\mathbf{W}'_i\mathbf{x}'_{i-1} + \mathbf{b}) - f(\mathbf{W}''_i\mathbf{x}''_{i-1} + \mathbf{b})\|_2$$

$$\leq \|\mathbf{x}'_{i-1} - \mathbf{x}''_{i-1}\|_2 + \frac{1}{N}\|\mathbf{W}'_i\mathbf{x}'_{i-1} - \mathbf{W}''_i\mathbf{x}''_{i-1}\|_2$$

$$\leq \|\mathbf{x}'_{i-1} - \mathbf{x}''_{i-1}\|_2 + \frac{1}{N}\|\mathbf{W}'_i\mathbf{x}'_{i-1} - \mathbf{W}'_i\mathbf{x}''_{i-1} + \mathbf{W}'_i\mathbf{x}''_{i-1} - \mathbf{W}''_i\mathbf{x}''_{i-1}\|_2$$

$$\leq \|\mathbf{x}'_{i-1} - \mathbf{x}''_{i-1}\|_2 + \frac{1}{N}\|\mathbf{W}'_i\mathbf{x}'_{i-1} - \mathbf{W}'_i\mathbf{x}''_{i-1}\|_2 + \frac{1}{N}\|\mathbf{W}'_i\mathbf{x}''_{i-1} - \mathbf{W}''_i\mathbf{x}''_{i-1}\|_2$$

$$\leq (1 + \frac{1}{N})\|\mathbf{x}'_{i-1} - \mathbf{x}''_{i-1}\|_2 + \frac{1}{N}\|\mathbf{x}''_{i-1}\|_2\|\mathbf{W}'_i - \mathbf{W}''_i\|_2$$

$$\leq (1 + \frac{1}{N})\|\mathbf{x}'_{i-1} - \mathbf{x}''_{i-1}\|_2 + \frac{1}{N}\left(e\|\mathbf{s}''\|_2 + (e-1)D_b\right)\|\mathbf{W}'_i - \mathbf{W}''_i\|_2$$

$$\leq (1 + \frac{1}{N})\|\mathbf{x}'_{i-1} - \mathbf{x}''_{i-1}\|_2$$
$$+ \frac{1}{N}\left(e\|\mathbf{s}''\|_2 + (e-1)D_b\right)\left((1 + \frac{1}{D})e^{4D^2} - \frac{1}{D}\right)\|\theta' - \theta''\|_2$$

By aggregating the last inequality for $i \in \{1, \ldots, N\}$ we conclude that

$$\|g_{\theta'}(\mathbf{s}'') - g_{\theta''}(\mathbf{s}'')\|_2 = \|\mathbf{\Omega}'_2\mathbf{x}'_N - \mathbf{\Omega}''_2\mathbf{x}''_N\|_2 = \|\mathbf{\Omega}'_2\mathbf{x}'_N - \mathbf{\Omega}'_2\mathbf{x}''_N + \mathbf{\Omega}'_2\mathbf{x}''_N - \mathbf{\Omega}''_2\mathbf{x}''_N\|_2$$
$$\leq \|\mathbf{\Omega}'_2\|_2\|\mathbf{x}'_N - \mathbf{x}''_N\|_2 + \|\mathbf{x}''_N\|_2\|\mathbf{\Omega}'_2 - \mathbf{\Omega}''_2\|_2$$
$$\leq \|\mathbf{x}'_N - \mathbf{x}''_N\|_2 + \left(e\|\mathbf{s}''\|_2 + (e-1)D_b\right)\|\theta' - \theta''\|_2$$
$$\leq (1 + \frac{1}{N})^N\|\mathbf{x}'_0 - \mathbf{x}''_0\|_2$$
$$+ \left(e\|\mathbf{s}''\|_2 + (e-1)D_b\right)\left(1 + \sum_{j=0}^{i-1}(1 + \frac{1}{N})^j \cdot \frac{1}{N}((1 + \frac{1}{D})e^{4D^2} - \frac{1}{D})\right)\|\theta' - \theta''\|_2$$
$$\leq (1 + \frac{1}{N})^N\|\mathbf{s}'' - \mathbf{s}''\|_2$$
$$+ \left(e\|\mathbf{s}''\|_2 + (e-1)D_b\right)\left(1 + ((1 + \frac{1}{N})^N - 1)((1 + \frac{1}{D})e^{4D^2} - \frac{1}{D})\right)\|\theta' - \theta''\|_2$$
$$\leq \left(e\|\mathbf{s}''\|_2 + (e-1)D_b\right)\left(1 + (e-1)((1 + \frac{1}{D})e^{4D^2} - \frac{1}{D})\right)\|\theta' - \theta''\|_2$$

The inequality above together with (20) and (21) concludes the proof of bound (18). $\qquad\square$

**Lemma 2.** *If Assumptions 4.2, 4.3 are satisfied, then for any $\theta' = \{\mathbf{\Omega}'_1, \mathbf{\Omega}'_2, \mathbf{b}', \mathbf{N}', \mathbf{Q}', \mathbf{W}'_0\} \in \mathbb{D}$ and $\theta'' = \{\mathbf{\Omega}''_1, \mathbf{\Omega}''_2, \mathbf{b}'', \mathbf{N}'', \mathbf{Q}'', \mathbf{W}''_0\} \in \mathbb{D}$ such that $\|\mathbf{N}'\|_2, \|\mathbf{Q}'\|_2, \|\mathbf{N}''\|_2, \|\mathbf{Q}''\|_2 \leq D, \|\mathbf{b}'\|_2, \|\mathbf{b}''\|_2 \leq D_b$ for some $D, D_b > 0$ it holds that*

$$|F(\theta') - F(\theta'')| \leq \mathcal{C}\|\theta' - \theta''\|_2$$

*where*

$$\mathcal{C} = L_2K((1+e)\gamma(K)L_1 + 1)\left(e(L_1^K(1+e)^K + 1)\|\mathbf{s}_0\|_2 + \gamma(K)\left(L_1((e-1)D_b + \|\mathbf{s}_0\|_2\right.\right.$$
$$\left.\left. + \|\widehat{\mathbf{a}}\|_2) + \|\text{env}^{(1)}(\mathbf{s}_0, \widehat{\mathbf{a}})\|_2\right) + (e-1)D_b\right)\left(1 + (e-1)((1 + \frac{1}{D})e^{4D^2} - \frac{1}{D})\right)\right)$$

$$\gamma(k) = \begin{cases} k, & \text{if } L_1(1+e) = 1 \\ \frac{L_1^k(1+e)^k - 1}{L_1(1+e) - 1}, & \text{otherwise} \end{cases}$$

*and $\widehat{\mathbf{a}}$ is an arbitrary fixed vector from $\mathbb{R}^m$.*

*Proof.* Let $\mathbf{s}'_1, l'_1 \ldots, \mathbf{s}'_K, l'_K$ and $\mathbf{s}''_0, l''_1 \ldots, \mathbf{s}''_K, l''_K$ be rollouts of (7) for $\theta'$ and $\theta''$ respectively starting from $\mathbf{s}'_0 = \mathbf{s}''_0 = \mathbf{s}_0$. In the light of Assumption 4.2 and Lemma 1 for any $k \in \{1, \ldots, K\}$ we have $\mathbf{s}''_k = \text{env}^{(1)}(\mathbf{s}''_{k-1}, g_{\theta''}(\mathbf{s}''_{k-1}))$ and, therefore,

$$\|\mathbf{s}''_k\|_2 = \|\text{env}^{(1)}(\mathbf{s}''_{k-1}, g_{\theta''}(\mathbf{s}''_{k-1})) - \text{env}^{(1)}(\mathbf{s}_0, \widehat{\mathbf{a}}) + \text{env}^{(1)}(\mathbf{s}_0, \widehat{\mathbf{a}})\|_2$$
$$\leq \|\text{env}^{(1)}(\mathbf{s}''_{k-1}, g_{\theta''}(\mathbf{s}''_{k-1})) - \text{env}^{(1)}(\mathbf{s}_0, \widehat{\mathbf{a}})\|_2 + \|\text{env}^{(1)}(\mathbf{s}_0, \widehat{\mathbf{a}})\|_2$$
$$\leq L_1\left(\|\mathbf{s}''_{k-1} - \mathbf{s}_0\|_2 + \|g_{\theta''}(\mathbf{s}''_{k-1}) - \widehat{\mathbf{a}}\|_2\right) + \|\text{env}^{(1)}(\mathbf{s}_0, \widehat{\mathbf{a}})\|_2$$
$$\leq L_1\|\mathbf{s}''_{k-1}\|_2 + L_1\|\mathbf{s}_0\|_2 + L_1\|g_{\theta''}(\mathbf{s}''_{k-1})\|_2 + L_1\|\widehat{\mathbf{a}}\|_2 + \|\text{env}^{(1)}(\mathbf{s}_0, \widehat{\mathbf{a}})\|_2$$
$$\leq L_1(1+e)\|\mathbf{s}''_{k-1}\|_2 + L_1((e-1)D_b + \|\mathbf{s}_0\|_2 + \|\widehat{\mathbf{a}}\|_2) + \|\text{env}^{(1)}(\mathbf{s}_0, \widehat{\mathbf{a}})\|_2 \leq \ldots$$

$$\leq L_1^k(1+e)^k\|\mathbf{s}_0\|_2 + \sum_{j=0}^{k-1} L_1^j(1+e)^j\left(L_1((e-1)D_b+\|\mathbf{s}_0\|_2+\|\widehat{\mathbf{a}}\|_2)+\|\mathrm{env}^{(1)}(\mathbf{s}_0,\widehat{\mathbf{a}})\|_2\right)$$

$$\leq L_1^k(1+e)^k\|\mathbf{s}_0\|_2 + \gamma(k)\left(L_1((e-1)D_b+\|\mathbf{s}_0\|_2+\|\widehat{\mathbf{a}}\|_2)+\|\mathrm{env}^{(1)}(\mathbf{s}_0,\widehat{\mathbf{a}})\|_2\right)$$

$$\leq L_1^K(1+e)^K\|\mathbf{s}_0\|_2 + \gamma(K)\left(L_1((e-1)D_b+\|\mathbf{s}_0\|_2+\|\widehat{\mathbf{a}}\|_2)+\|\mathrm{env}^{(1)}(\mathbf{s}_0,\widehat{\mathbf{a}})\|_2\right)$$

$$= L_1^K(1+e)^K\|\mathbf{s}_0\|_2 + \gamma(K)\mathcal{A}$$

where we denote

$$\mathcal{A} = L_1((e-1)D_b+\|\mathbf{s}_0\|_2+\|\widehat{\mathbf{a}}\|_2)+\|\mathrm{env}^{(1)}(\mathbf{s}_0,\widehat{\mathbf{a}})\|_2.$$

In addition to that, denote

$$\mathcal{B} = 1+(e-1)((1+\frac{1}{D})e^{4D^2}-\frac{1}{D}).$$

From the last inequality it follows that

$$\|\mathbf{s}_{k-1}''\|_2 \leq \max(\|\mathbf{s}_0\|_2, L_1^K(1+e)^K\|\mathbf{s}_0\|_2+\gamma(K)\mathcal{A}) \leq (L_1^K(1+e)^K+1)\|\mathbf{s}_0\|_2+\gamma(K)\mathcal{A}$$

Next, observe that

$$\|\mathbf{s}_k'-\mathbf{s}_k''\|_2 = \|\mathrm{env}^{(1)}(\mathbf{s}_{k-1}',g_{\theta'}(\mathbf{s}_{k-1}')) - \mathrm{env}^{(1)}(\mathbf{s}_{k-1}'',g_{\theta''}(\mathbf{s}_{k-1}''))\|_2$$

$$\leq L_1\left(\|\mathbf{s}_{k-1}'-\mathbf{s}_{k-1}''\|_2+\|g_{\theta'}(\mathbf{s}_{k-1}')-g_{\theta''}(\mathbf{s}_{k-1}'')\|_2\right)$$

$$\leq L_1(1+e)\|\mathbf{s}_{k-1}'-\mathbf{s}_{k-1}''\|_2$$
$$+L_1\left(e\|\mathbf{s}_{k-1}''\|_2+(e-1)D_b\right)\left(1+(e-1)((1+\frac{1}{D})e^{4D^2}-\frac{1}{D})\right)\|\theta'-\theta''\|_2$$

$$\leq L_1(1+e)\|\mathbf{s}_{k-1}'-\mathbf{s}_{k-1}''\|_2 + L_1\left(e((L_1^K(1+e)^K+1)\|\mathbf{s}_0\|_2+\gamma(K)\mathcal{A})\right.$$

$$\left.+(e-1)D_b\right)\mathcal{B}\|\theta'-\theta''\|_2 \leq \ldots$$

$$\leq L_1^k(1+e)^k\|\mathbf{s}_0-\mathbf{s}_0\|_2 + \sum_{j=0}^{k-1}L_1^j(1+e)^j L_1\left(e((L_1^K(1+e)^K+1)\|\mathbf{s}_0\|_2+\gamma(K)\mathcal{A})\right.$$

$$\left.+(e-1)D_b\right)\mathcal{B}\|\theta'-\theta''\|_2$$

$$= \gamma(k)L_1\left(e((L_1^K(1+e)^K+1)\|\mathbf{s}_0\|_2+\gamma(K)\mathcal{A})+(e-1)D_b\right)\mathcal{B}\|\theta'-\theta''\|_2$$

$$\leq \gamma(K)L_1\left(e((L_1^K(1+e)^K+1)\|\mathbf{s}_0\|_2+\gamma(K)\mathcal{A})+(e-1)D_b\right)\mathcal{B}\|\theta'-\theta''\|_2.$$

We conclude by deriving that

$$|F(\theta')-F(\theta'')| \leq \sum_{k=1}^K |l_k'-l_k''| \leq \sum_{k=1}^K |\mathrm{env}^{(2)}(\mathbf{s}_{k-1}',g_{\theta'}(\mathbf{s}_{k-1}')) - \mathrm{env}^{(2)}(\mathbf{s}_{k-1}'',g_{\theta''}(\mathbf{s}_{k-1}''))|$$

$$\leq \sum_{k=1}^K L_2\left(\|\mathbf{s}_{k-1}'-\mathbf{s}_{k-1}''\|_2+\|g_{\theta'}(\mathbf{s}_{k-1}')-g_{\theta''}(\mathbf{s}_{k-1}'')\|_2\right)$$

$$\leq L_2\sum_{k=1}^K\left((1+e)\|\mathbf{s}_{k-1}'-\mathbf{s}_{k-1}''\|_2 + \left(e\|\mathbf{s}_{k-1}''\|_2+(e-1)D_b\right)\mathcal{B}\|\theta'-\theta''\|_2\right)$$

$$\leq L_2\sum_{k=1}^K\left((1+e)\|\mathbf{s}_{k-1}'-\mathbf{s}_{k-1}''\|_2\right.$$

$$+ \left( e(L_1^K(1+e)^K + 1)\|\mathbf{s}_0\|_2 + \gamma(K)\mathcal{A} + (e-1)D_b \right) \mathcal{B}\|\theta' - \theta''\|_2 \right)$$

$$\leq L_2 \sum_{k=1}^{K} \left( (1+e)\gamma(K)L_1 \left( e((L_1^K(1+e)^K + 1)\|\mathbf{s}_0\|_2 + \gamma(K)\mathcal{A}) + (e-1)D_b \right) \right.$$

$$\times \mathcal{B}\|\theta' - \theta''\|_2 + \left( e(L_1^K(1+e)^K + 1)\|\mathbf{s}_0\|_2 + \gamma(K)\mathcal{A} + (e-1)D_b \right) \mathcal{B}\|\theta' - \theta''\|_2 \right)$$

$$= L_2 K((1+e)\gamma(K)L_1 + 1) \left( e(L_1^K(1+e)^K + 1)\|\mathbf{s}_0\|_2 + \gamma(K)\mathcal{A} \right.$$

$$+ (e-1)D_b \right) \mathcal{B}\|\theta' - \theta''\|_2$$

$$\square$$

**Lemma 3.** *If Assumptions 4.2, 4.3 are satisfied, then for any* $\theta' = \{\mathbf{\Omega}_1', \mathbf{\Omega}_2', \mathbf{b}', \mathbf{N}', \mathbf{Q}', \mathbf{W}_0'\} \in \mathbb{D}$ *and* $\theta'' = \{\mathbf{\Omega}_1'', \mathbf{\Omega}_2'', \mathbf{b}'', \mathbf{N}'', \mathbf{Q}'', \mathbf{W}_0''\} \in \mathbb{D}$ *such that* $\|\mathbf{N}'\|_2, \|\mathbf{Q}'\|_2, \|\mathbf{N}''\|_2, \|\mathbf{Q}''\|_2 \leq D, \|\mathbf{b}'\|_2, \|\mathbf{b}''\|_2 \leq D_b$ *for some* $D, D_b > 0$ *it holds that*

$$\|\nabla F_\sigma(\theta') - \nabla F_\sigma(\theta'')\|_2 \leq \frac{\mathcal{C}\sqrt{l}}{\sigma}\|\theta' - \theta''\|_2$$

*where* $\mathcal{C}$ *is from the definition of Lemma 2.*

*Proof.* We deduce that

$$\|\nabla F_\sigma(\theta') - \nabla F_\sigma(\theta'')\|_2^2 = \frac{1}{\sigma^2}\|\mathbb{E}_{\epsilon \sim \mathcal{N}(0,I)}(F(\theta' + \sigma\epsilon) - F(\theta' + \sigma\epsilon))\epsilon\|_2^2$$

$$\leq \frac{1}{\sigma^2}\mathbb{E}_{\epsilon \sim \mathcal{N}(0,I)}\|(F(\theta' + \sigma\epsilon) - F(\theta' + \sigma\epsilon))\epsilon\|_2^2$$

$$= \frac{1}{\sigma^2}\mathbb{E}_{\epsilon \sim \mathcal{N}(0,I)}(F(\theta' + \sigma\epsilon) - F(\theta' + \sigma\epsilon))^2\|\epsilon\|_2^2$$

$$\leq \frac{1}{\sigma^2}\mathbb{E}_{\epsilon \sim \mathcal{N}(0,I)}\mathcal{C}^2\|\theta' - \theta''\|_2^2\|\epsilon\|_2^2$$

$$= \frac{\mathcal{C}^2}{\sigma^2}\|\theta' - \theta''\|_2^2 \cdot \mathbb{E}_{\epsilon \sim \mathcal{N}(0,I)}\|\epsilon\|_2^2 = \frac{\mathcal{C}^2 \cdot l}{\sigma^2}\|\theta' - \theta''\|_2^2.$$

$$\square$$

**Lemma 4.** *If Assumption 4.2 is satisfied, then for any* $\theta \in \mathbb{D}$

$$\mathbb{E}\|\widetilde{\nabla} F_\sigma(\theta)\|_2^2 \leq \frac{K^2 M^2 l}{\sigma^2}.$$

*Proof.* By using Assumption 4.2 and that $|F(\theta)| = |\sum_{k=1}^{K} l_k| \leq KM$ we derive

$$(\mathbb{E}\|\widetilde{\nabla} F_\sigma(\theta)\|_2)^2 \leq \mathbb{E}\|\widetilde{\nabla}_\theta F_\sigma(\theta)\|_2^2 = \frac{1}{v^2\sigma^2}\mathbb{E}_{\{\epsilon_w \sim \mathcal{N}(0,I)\}}\|\sum_{w=1}^{v} F(\theta + \sigma\epsilon_w)\epsilon_w\|_2^2$$

$$\leq \frac{1}{v\sigma^2}\mathbb{E}_{\{\epsilon_w \sim \mathcal{N}(0,I)\}}\sum_{w=1}^{v}\|F(\theta + \sigma\epsilon_w)\epsilon_w\|_2^2$$

$$= \frac{1}{\sigma^2}\mathbb{E}_{\epsilon \sim \mathcal{N}(0,I)}\|F(\theta + \sigma\epsilon)\epsilon\|_2^2 = \frac{1}{\sigma^2}\mathbb{E}_{\epsilon \sim \mathcal{N}(0,I)}F(\theta + \sigma\epsilon)^2\|\epsilon\|_2^2$$

$$\leq \frac{1}{\sigma^2}\mathbb{E}_{\epsilon \sim \mathcal{N}(0,I)}K^2 M^2\|\epsilon\|_2^2 = \frac{K^2 M^2}{\sigma^2}\mathbb{E}_{\epsilon \sim \mathcal{N}(0,I)}\|\epsilon\|_2^2 = \frac{K^2 M^2 l}{\sigma^2}$$

$$\square$$

*Proof of Theorem 1.* Hereafter we assume that $\mathcal{F}_{\tau,D,D_b}$ holds for random iterates $\theta^{(0)}, \ldots, \theta^{(\tau)}$. According to Lemma 3, $\frac{\mathcal{C}\sqrt{l}}{\sigma}$ is a bound on $F_\sigma(\theta)$'s Hessian on $\{\theta \in \mathbb{D} \mid \|\mathbf{N}\|_2, \|\mathbf{Q}\|_2 < D, \mathbf{b} < D_b\}$. We apply [6, Appendix] to derive that for any $\tau' \le \tau$

$$F(\theta^{(\tau')}) - F(\theta^{(\tau'-1)}) \ge \alpha_{\tau'} \nabla_{\mathcal{R}} F_\sigma(\theta^{(\tau'-1)})^\top \widetilde{\nabla}_{\mathcal{R}} F_\sigma(\theta^{(\tau'-1)}) - \frac{\mathcal{C}\sqrt{l}}{2\sigma} \alpha_{\tau'}^2 \|\widetilde{\nabla}_{\mathcal{R}} F_\sigma(\theta^{(\tau'-1)})\|_F^2 \tag{23}$$

Let $\mathcal{A}$ denote a sigma-algebra associated with $\theta^{(1)}, \ldots, \theta^{(\tau'-1)}$. We use that $\mathbb{E}[\widetilde{\nabla}_{\mathcal{R}} F_\sigma(\theta^{(\tau'-1)})|\mathcal{A}] = \nabla_{\mathcal{R}} F_\sigma(\theta^{(\tau'-1)})$ and $\mathbb{E}[\|\widetilde{\nabla}_{\mathcal{R}} F_\sigma(\theta^{(\tau'-1)})\|_2^2|\mathcal{A}] \le \frac{K^2 M^2 l}{\sigma^2}$ (Lemma 4) and take an expectation of (21) conditioned on $\mathcal{A}$:

$$\mathbb{E}[F(\theta^{(\tau')})|\mathcal{A}] - F(\theta^{(\tau'-1)}) \ge \alpha_{\tau'} \|\nabla_{\mathcal{R}} F_\sigma(\theta^{(\tau'-1)})\|_2^2 - \frac{\mathcal{C}\sqrt{l}}{2\sigma} \alpha_{\tau'}^2 \cdot \frac{K^2 M^2 l}{\sigma^2}.$$

Regroup the last inequality and take a full expectation to obtain

$$\alpha_{\tau'} \mathbb{E}\|\nabla_{\mathcal{R}} F_\sigma(\theta^{(\tau'-1)})\|_2^2 \le \mathbb{E} F(\theta^{(\tau')}) - \mathbb{E} F(\theta^{(\tau'-1)}) + \frac{\mathcal{C}\sqrt{l}}{2\sigma} \alpha_{\tau'}^2 \cdot \frac{K^2 M^2 l}{\sigma^2}.$$

Perform a summation of the last inequality for all $\tau' \in \{1, \ldots, \tau\}$:

$$\sum_{\tau'=1}^{\tau} \alpha_{\tau'} \mathbb{E}\|\nabla_{\mathcal{R}} F_\sigma(\theta^{(\tau'-1)})\|_2^2 \le \mathbb{E} F(\theta^{(\tau)}) - F(\theta^{(0)}) + \frac{\mathcal{C} K^2 M^2 l^{\frac{3}{2}}}{2\sigma^3} \sum_{\tau'=1}^{\tau} \alpha_{\tau'}^2$$

$$\le KM - F(\theta^{(0)}) + \frac{\mathcal{C} K^2 M^2 l^{\frac{3}{2}}}{2\sigma^3} \sum_{\tau'=1}^{\tau} \alpha_{\tau'}^2$$

since $F(\theta) \le KM$ for all $\theta \in \mathbb{D}$. We next observe that

$$\min_{0 \le \tau' < \tau} \mathbb{E}\|\nabla_{\mathcal{R}} F_\sigma(\theta^{(\tau')})\|_2^2 = \frac{\sum_{\tau'=1}^{\tau} \alpha_{\tau'} \min_{0 < \tau' \le \tau} \mathbb{E}\|\nabla_{\mathcal{R}} F_\sigma(\theta^{(\tau'-1)})\|_2^2}{\sum_{\tau'=1}^{\tau} \alpha_{\tau'}}$$

$$\le \frac{1}{\sum_{\tau'=1}^{\tau} \alpha_{\tau'}} (KM - F(\theta^{(0)})) + \frac{\sum_{\tau'=1}^{\tau} \alpha_{\tau'}^2}{\sum_{\tau'=1}^{\tau} \alpha_{\tau'}} \cdot \frac{\mathcal{C} K^2 M^2 l^{\frac{3}{2}}}{2\sigma^3}.$$

The proof is concluded by observing that $\sum_{\tau'=1}^{\tau} \alpha_{\tau'} = \Omega(\tau^{0.5})$ and $\sum_{\tau'=1}^{\tau} \alpha_{\tau'}^2 = O(\log \tau) = o(\tau^\epsilon)$ for any $\epsilon > 0$.

$\square$