[Reviews · NeurIPS 2020]

Review 1

Summary and Contributions: Post Rebuttal: Authors have addressed my concerns. And in light of this I am increasing the score. Here're my brief responses. (1) ANODEV2: The choice of calling the method of [56] (which is closely related to the present paper) HyperNets and not ANODEV2 was curious, to say nothing of being confusing. I had previously assumed that by HyperNets authors had meant the method of [26] even though they cite [56] at one place and [26] at another place. I have taken authors' claim on face value. (2) I retain my reservations about the theoretical result. The paper provides methods to constrain the parameter matrix of (coupled) neural ODEs to be orthogonal. It shows theoretically that doing so alleviates the exploding-vanishing gradient problem often associated with training of deep models. It also shows experimental results for reinforcement and supervised learning.

Strengths: The paper seems to fit the template of let's apply (pre-existing) idea A that's been used in setting B to a new variant setting C. In this case, A=orthogonality constraints on the parameters, B=deep/recurrent neural network training, C=Neural ODE (more precisely coupled neural ODE as in ANODEV2). However, the application of A to C is not trivial and requires transporting methods from other areas of mathematics (differential equations, optimization on manifolds, flows on manifolds) and some expertise seems to be necessary to carry this out. Lemma 4.1 which proves that gradients do not vanish or explode also seems interesting. The method seems to do well in experiments.

Weaknesses: While orthogonality constraint helps with vanishing-exploding gradients issue as shown in Lemma 4.1, it's not clear to me if it doesn't introduce some new issues. For example, by reducing the flexibility in how W can change, training might become slow or not converge to a point with near-minimum loss. Theorem 1 is proving convergence to a stationary point which of course need not have small loss. It's worth noting that Lemma 4.1 doesn't get used (as far as I can tell) in any other theoretical result. Thus, at least theoretically, the results in the paper can't be regarded as providing support for orthogonality constraints. I would like to know what the authors think about these issues. (The remark on line 34 seems to be related: "Fortunately, there exist several efficient parameterizations of the subgroups of the orthogonal group O(d) that, even though in principle reduce representational capacity, in practice produce high-quality models and bypass Riemannian optimization [36, 40, 34].") Experiments: The paper should justify the choices of baselines. The experiments could have included results with ANODEV2 (code is available online) which is closely related to the present paper, and possibly other recent neural ODE work (e.g. augmented NODE). Apart from BaseODE, another NODE benchmark chosen in the paper is NANODE. I had a quick look at that paper, and that also doesn't seem to compare with ANODEV2. In supervised learning, for neural net models used as baseline instead of just fully-connected feedforward networks one would also like to see ResNets (which are used as baseline for reinforcement learning experiments). I am not qualified to judge the importance of improvements to Evolution Strategies method for reinforcement learning as shown in this paper within the broader landscape of reinforcement learning methods.

Correctness: While I didn't verify the proofs in detail, I did not find any inaccuracies.

Clarity: The paper is reasonably well-written.

Relation to Prior Work: There's a good discussion of prior work. One thing I would like to know and is not discussed: is there a counterpart of Lemma 4.1 for neural network training in the previous literature.

Reproducibility: No

Additional Feedback: "Such Neural ODE constructions enable deeper models than would not otherwise be possible with a 27 fixed computation budget; however, it has been noted that training instabilities and the problem of 28 vanishing/exploding gradients can arise during the learning of very deep systems [43, 4, 23]." Unless I misunderstood, this whole sentence should be rewritten. In particular part of it should probably read "…than would otherwise be possible…" or ""…that would not otherwise be possible…" The second part of the sentence seems to support the first but "however" suggests otherwise.


Review 2

Summary and Contributions: This paper presents ODEtoODE, a new version of Neural ODE where not only the state vector x but also the parameter vector W evolve according to ODEs (Eq.2) respectively. A key idea is to restrict the parameters in the space of a compact matrix manifold \Sigma. As a result, in Lemma 4.1, the proposed method can avoid the vanishing/exploding gradient problem when \Sigma is the orthogonal group O(d). (It would be more convincing if this is empirically visualized in the Experiments section.) The trainable parameters are not W itself but the \psi in a map b_\psi that eats point W and spits out a tangent vector on \Sigma at W. The authors propose two implementations of b_\psi: ISO-ODEtoODE and Gated-ODEtoODE. In Theorem 1, the authors claim that the training procedure of the proposed method can ‘strongly converge’ in a certain reinforcement learning setup. (To be exact, they show the norm convergence of gradients.) The Experiments section lists several SOTA results on reinforcement learning.

Strengths: The idea to restrict parameters to O(d) is simple but tractable and effective both in theory (Lemma 4.1) and practice (Experiments).

Weaknesses: It would be more convincing if the gradients during training are visualized in the Experiments section.

Correctness: Correct. I found some flaws in mathematical notions. For examples, between ll.16-17, 'a solution' it is not a solution but an integral equation. At ll.10, 54 and 149 the authors assert 'strong convergence' but Theorem 1 only shows the norm convergence of the gradients.

Clarity: Average. Considering the simplicity of the main idea, Sections 1-4 could be more straight, simpler and clearer. For example, at the end of the Introduction, the authors summarize the contributions but I could not make sense what ISO/Gated-ODEtoODE are, what they solved, and how. At l.102 the authors introduce ‘a parameterized function b_\psi...’ without mentioning what the parameters are.

Relation to Prior Work: Not exhaustive, but covers wide viewpoints. Sometimes citations are rude, for example, [23] at l.28 and [35] at l.104 because these are books. I would like to read more detailed discussions in Section 5.2.3 as this is an empirical study.

Reproducibility: No

Additional Feedback: === UPDATE AFTER AUTHOR FEEDBACK === I have read the author’s responses and other reviewers’ comments, and I would keep my score. I do not agree with that the authors claim that HyperNets is ANODEV2. It is simply missed in the experiments.


Review 3

Summary and Contributions: This paper proposes a neural ODE framework that allows its parameters to be dynamically evolved through time. The paper also theoretically show that the proposed method, called ODEtoODE, can achieve a stable and effective training by constraining the parameter-flow on compact manifolds. The proposed method is evaluated for Reinforcement Learning tasks.

Strengths: - The topic is of interest to the research community. NeuralODE is a relatively new topic and is drawing more and more attentions recently. - The contribution is novel. Formulating parameter-flow and constraining it with orthogonal groups could be very helpful to understanding NeuralODE.

Weaknesses: - The paper is a little bit hard to follow. There are plenty of contents discussing the validity of ODEtoODE but have little paragraphs on how to apply the proposed method into usage. Since parameters become dynamical, it is expected to train ODEtoODE with additional procedures and may be different from the training of conventional ones. So including an algorithmic description is a good option. - Although the paper gives certain analysis on how ODEtoODE can help alleviate gradient vanishing/explosion problem, there is little experimental result to support such claim, especially for supervised learning in section 5.2. Without results such as convergence curves, etc, It is less convincing that the proposed method could really improve the stability and effectiveness.

Correctness: The proposed method technically sounds. And empirical settings look valid. However, certain experimental results are missing. Please refer to the weakness section.

Clarity: The technical details may need some tuning with more elaborations on how to apply the proposed method to solve problems.

Relation to Prior Work: Yes.

Reproducibility: Yes

Additional Feedback: --------------- After rebuttal ---------------- I decide to keep my score unchanged after reading other reviews and rebuttal.

[Author Response · NeurIPS 2020]

We would like to sincerely thank all the reviewers for reading the paper carefully and their very valuable feedback.
**General comments:**
**Visualization of vanishing/exploding gradients:** Similarly, as in Fig. 4 from [2], we plotted $L_2$-norms of the gradients
$\frac{\partial \mathcal{L}}{\partial \mathbf{x}_t}$ as a function of time $t$ at the beginning of the optimization and after 100 iterations (first two plots in Fig. 1). We
see that the norm of the gradient for ODEtoODE barely changes, while for the NeuralODE it converges to 0 as we
backpropagate through time (we observed convergence to 0 also for other methods, yet for the clarify of the picture we
did not present additional curves). The plots were created for Humanoid training from the paper, where ODEtoODE
was **the only method** that successfully trained the agent.
**Presentation:** in the final version we will simplify Sections 1-4 and provide definitions of ISO/Gated-ODEtoODE
in the Introduction. As suggested, we will improve citations' style. In Section 5.2.3, we will provide more detailed
discussion, and incorporate new results (see: Table from Fig. 1). Training ODEtoODEs does not require any additional
modules since the parameters governing the evolution of weight matrices are unconstrained and therefore can be handled
by standard backpropagation in the supervised setting and standard GD-approach with ES-gradients for the RL one. We
do agree that this requires clarification, thus in the final version we will incorporate an algorithmic box on ODEtoODEs
in the experimental section for clarity.
**Reviewer 1:**
**ANODEV2:** Thank you for pointing this out. The HyperNets networks we compare against in Sec. 5.2 (see: Table 1,
2) are precisely ANODEV2 architectures (as indicated by our citation [56] to ANODEV2 in l.265). This is also the case
for the ES experiments, and thus citation [26] (which is a typo) in l.238 of Sec. 5.1 should be replaced by [56].
**Pros & cons of orthogonality:**
Even though in principle deep architectures leveraging orthogonal matrices can hurt accuracy by restricting model
capacity, a rich line of work ([2,30,40]) on orthogonal RNNs and related methods shows that they do not, if designed
correctly. In fact, the main motivation behind these architectures are accuracy improvements due to training not suffering
from vanishing/exploding gradients - one of the critical problems in training deep machine learning systems. We also
demonstrate it exhaustively in the empirical section, covering both: the supervised and RL setup (the latter usually
not touched in the papers on the subject). To summarize, ODEtoODEs outperform accuracy-wise other methods (in
particular highly competitive ANODEV2) in **11 out of 15** considered supervised tasks, as we clearly state in the paper
(l.272). In the RL setting ODEtoODE is **the only method** that manages to train the most difficult Humanoid task and,
as we demonstrate in "*General comments*" section above, it is precisely due to its ability to stabilize gradients' lengths.
**Theoretical results:** We are not aware of **any** results in the literature on neural networks (even in the simpler setting with
shallow discrete-architectures & supervised learning) on the convergence to global minimum under weak assumptions
from our Theorem 1 and to the best of our knowledge we are the first to show convergence to local minima with
depth-independent bounds in the more challenging ES scenario. Given that, we interpret our theoretical results as the
strength of the paper. Lemma 4.1 is implicitly used in the proof of our Theorem 1 (see: Eq. 21 in the Appendix) and
techniques used to prove it (Sec. 8) play key role in establishing depth-independent bounds. In the final version of
the paper we will explicitly refer to it. Lemma 4.1. is also a natural continuous extension of the analogous results
for the orthogonal RNNs (see for instance: [2]), so can be used in various analogous continuous variants, not only
ODEtoODEs. We will clarify it in the final version.
**ResNets in supervised setting:** We run these additional experiments and present results in Table from Fig. 1.
**Reviewer 3:**
Thank you very much for the review.
**Mathematical notation:** we will incorporate suggested improvements and clarify in the final version (in particular, by
"strong convergence results/guarantees" we mean "compelling"). **L.102:** This is a generic notation with parameters
encapsulated in $\psi$. We will clarify this in the final version.
**Visualization of gradients' norms:** See, our paragraph on visualization in "*General comments*" section.
**Reviewer 4:**
Thank you very much for the review.
**Visualization of gradients' norms:** See, our paragraph on visualization in "*General comments*" section.

| Tasks | Dotted Lines | Spatter | Stripe | Translate | Rotate | Scale | Shear | Motion Blur | Glass Blur | Shot Noise | Identity |
|---|---|---|---|---|---|---|---|---|---|---|---|
| ResNet-1 | 92.89 | 93.6 | 30.06 | 25.97 | 82.59 | 58.11 | 91.86 | 76.55 | 91.22 | 96.15 | 97.65 |
| ResNet-10 | 92.9 | 93.56 | 27.96 | 26.1 | 83.25 | 63.56 | 91.44 | 72.42 | 89.16 | 94.31 | 97.38 |
| ODEtoODE-1 | **95.42** | **94.9** | **44.51** | **26.82** | 83.9 | 66.68 | **93.37** | **78.63** | **93.91** | **96.91** | **97.94** |
| ODEtoODE-10 | 95.22 | **94.9** | 44.37 | 26.63 | **84.1** | **66.76** | 92.93 | 77.58 | 93.29 | 96.71 | 97.88 |

Figure 1: Showcasing gradient vanishing & extra experiments with ResNets. First two plots present norm of the gradient $\frac{\partial \mathcal{L}}{\partial \mathbf{x}_t}$ as a function of $t$ for Humanoid training after 0 (first plot) and 100 iterations (second plot). Time = 1000 corresponds to $T = 1.0$ from the paper. Table presents additional results on ResNets and comparison with ODEtoODEs. **Bolded blue** correspond to best results and **they are all** for variants of ODEtoODEs.

[Meta-Review · NeurIPS 2020]

The paper consider a particular class of neural ODEs and shows that constraining the flow to lie in O(d) improves the stability of training. Initially, the paper received mixed reviews (1 below threshold 2 above threshold). On the positive side, the contribution is found interesting and novel, the idea of restricting to a compact manifold to be tractable and effective. On the negative side, the reviewers had some issues with technical details, would have liked the paper to be a little bit more clear, and would have liked more experiments. The reviewers were satisfied by the rebuttal: R1 writes he/she will increase his/her rating, with the other two reviewers remained positive. In the end, all three reviewers agree the paper is above threshold.